# POWER UP! ROBUST GRAPH CONVOLUTIONAL NETWORK BASED ON GRAPH POWERING

## ABSTRACT

Graph convolutional networks (GCNs) are powerful tools for graph-structured data. However, they have been recently shown to be vulnerable to topological attacks. To enhance adversarial robustness, we go beyond spectral graph theory to robust graph theory. By challenging the classical graph Laplacian, we propose a new convolution operator that is provably robust in the spectral domain and is incorporated in the GCN architecture to improve expressivity and interpretability. By extending the original graph to a sequence of graphs, we also propose a robust training paradigm that encourages transferability across graphs that span a range of spatial and spectral characteristics. The proposed approaches are demonstrated in extensive experiments to simultaneously improve performance in both benign and adversarial situations.

## 1 INTRODUCTION

Graph convolutional networks (GCNs) are powerful extensions of convolutional neural networks (CNN) to graph-structured data. Recently, GCNs and variants have been applied to a wide range of domains, achieving state-of-the-art performances in social networks (Kipf & Welling, 2017), traffic prediction (Rahimi et al., 2018), recommendation systems (Ying et al., 2018), applied chemistry and biology (Kearnes et al., 2016; Fout et al., 2017), and natural language processing (Atwood & Towsley, 2016; Hamilton et al., 2017; Bastings et al., 2017; Marcheggiani & Titov, 2017), just to name a few (Zhou et al., 2018; Wu et al., 2019).

GCNs belong to a family of *spectral methods* that deal with spectral representations of graphs (Zhou et al., 2018; Wu et al., 2019). A fundamental ingredient of GCNs is the graph convolution operation defined by the graph Laplacian in the Fourier domain:

$$g_{\theta} \star x := \hat{g}_{\theta}(L)x, \tag{1}$$

where $x \in \mathbb{R}^n$ is the graph signal on the set of vertices $\mathcal{V}$ and $\hat{g}_{\theta}$ is a spectral function applied to the graph Laplacian $L := D - A$ (where $D$ and $A$ are the degree matrix and the adjacency matrix, respectively). Because this operation is computational intensive for large graphs and non-spatially localized (Bruna et al., 2014), early attempts relied on a parameterization with smooth coefficients (Henaff et al., 2015) or a truncated expansion in terms of of Chebyshev polynomials (Hammond et al., 2011). By further restricting the Chebyshev polynomial order by 2, the approach in (Kipf & Welling, 2017) referred henceforth as the vanilla GCN pushed the state-of-the-art performance of semi-supervised learning. The network has the following layer-wise update rule:

$$H^{(l+1)} := \psi\left(\mathcal{A}H^{(l)}W^{(l)}\right), \tag{2}$$

where $H^{(l)}$ is the $l$-th layer hidden state (with $H^{(1)} := X$ as nodal features), $W^{(l)}$ is the $l$-th layer weight matrix, $\psi$ is the usual point-wise activation function, and $\mathcal{A}$ is the convolution operator chosen to be the degree weighted Laplacian with some slight modifications (Kipf & Welling, 2017). Subsequent GCN variants have different architectures, but they all share the use of the Laplacian matrix as the convolution operator (Zhou et al., 2018; Wu et al., 2019).

### 1.1 WHY NOT GRAPH LAPLACIAN?

Undoubtedly, the Laplacian operator (and its variants, e.g., normalized/powered Laplacian) plays a central role in spectral theory, and is a natural choice for a variety of spectral algorithms such as

principal component analysis, clustering and linear embeddings (Chung & Graham, 1997; Belkin & Niyogi, 2002). *So what can be problematic?*

From a spatial perspective, GCNs with $d$ layers cannot acquire nodal information beyond its $d$-distance neighbors; hence, it severely limits its scope of data fusion. Recent works (Lee et al., 2018; Abu-El-Haija et al., 2018; 2019; Wu et al., 2019) alleviated this issue by directly powering the graph Laplacian.

From a spectral perspective, one could demand better *spectral properties*, given that GCN is fundamentally a particular (yet effective) approximation of the spectral convolution (1). A key desirable property for generic spectral methods is known as "spectral separation," namely the spectrum should comprise a few dominant eigenvalues whose associated eigenvectors reveal the sought structure in the graph. A well-known prototype is the Ramanujan property, for which the second leading eigenvalue of a $r$-regular graph is no larger than $2\sqrt{r-1}$, which is also enjoyed asymptotically by random $r$-regular graphs (Friedman, 2004) and Erdős-Rényi graphs that are not too sparse (Feige & Ofek, 2005). In a more realistic scenario, consider the stochastic block model (SBM), which attempts to capture the essence of many networks, including social graphs, citation graphs, and even brain networks (Holland et al., 1983).

**Definition 1** (Simplified stochastic block model)**.** The graph $\mathcal{G}$ with $n$ nodes is drawn under $\mathrm{SBM}(n, k, a_{\mathrm{intra}}, a_{\mathrm{inter}})$ if the nodes are evenly and randomly partitioned into $k$ communities, and nodes $i$ and $j$ are connected with probability $a_{\mathrm{intra}}/n \in [0, 1]$ if they belong to the same community, and $a_{\mathrm{inter}}/n \in [0, 1]$ if they are from different communities.

It turns out that for community detection, the top $k$ leading eigenvectors of the adjacency matrix $\boldsymbol{A}$ play an important role. In particular, for the case of 2 communities, spectral bisection algorithms simply take the second eigenvector to reveal the community structure. This can be also seen from the expected adjacency matrix $\mathbb{E}[\boldsymbol{A}]$ under $\mathrm{SBM}(n, 2, a_{\mathrm{intra}}, a_{\mathrm{inter}})$, which is a rank-2 matrix with the top eigenvalue $\frac{1}{2}(a_{\mathrm{intra}} + a_{\mathrm{inter}})$ and eigenvector $\mathbf{1}$, and the second eigenvalue $\frac{1}{2}(a_{\mathrm{intra}} - a_{\mathrm{inter}})$ and eigenvector $\boldsymbol{\sigma}$ such that $\sigma_i = 1$ if $i$ is in community 1 and $\sigma_i = -1$ otherwise. More generally, the second eigenvalue is of particular theoretical interests because it controls at the first order how fast heat diffuses through graph, as depicted by the discrete Cheeger inequality (Lee et al., 2014).

While one would expect taking the second eigenvector of the adjacency matrix suffices, it often fails in practice (even when it is theoretically possible to recover the clusters given the signal-to-noise ratio). This is especially true for sparse networks, whose average nodal degrees is a constant that does not grow with the network size. This is because the spectrum of the Laplacian or adjacency matrix is blurred by "outlier" eigenvalues in the sparse regime, which is often caused by high degree nodes (Kaufmann et al., 2016). Unsurprisingly, powering the Laplacian would be of no avail, because it does not change the eigenvectors or the ordering of eigenvalues. In fact, those outliers can become more salient after powering, thereby weakening the useful spectral signal even further. Besides, pruning the largest degree nodes in the adjacency matrix or normalizing the Laplacian cannot solve the issue. To date, the best results for pruning does not apply down to the theoretical recovery threshold (Coja-Oghlan, 2010; Mossel et al., 2012; Le et al., 2015); either outliers would persist or one could prune too much that the graph is destroyed. As for normalized Laplacian, it may overcorrect the large degree nodes, such that the leading eigenvectors would catch the "tails" of the graph, i.e., components weakly connected to the main graph. See Figure A.3 in the appendix for an experimental illustration.

**In summary, graph Laplacian may not be the ideal choice due to its limited spatial scope of information fusion, and its undesirable artefacts in the spectral domain.**

### 1.2 IF NOT LAPLACIAN, THEN WHAT?

In searching for alternatives, potential choices are many, so it is necessary to clarify the goals. In view of the aforementioned pitfalls of the graph Laplacian, one would naturally ask the question:

**Can we find an operator that has wider spatial scope, more robust spectral properties, and is meanwhile interpretable and can increase the expressive power of GCNs?**

From a perspective of *graph data analytics*, this question gauges how information is propagated and fused on a graph, and how we should interpret "adjacency" in a much broader sense. An image can

be viewed as a regular grid, yet the operation of a CNN filter goes beyond the nearest pixel to a local neighborhood to extract useful features. How to draw an analogy to graphs?

From a perspective of *robust learning*, this question sheds light on the basic observation that real-world graphs are often noisy and even adversarial. The nice spectral properties of a graph topology can be lost with the presence or absence of edges. What are some principled ways to robustify the convolution operator and graph embeddings?

In this paper, we propose a graph learning paradigm that aims at achieving this goal, as illustrated in Figure 1. The key idea is *to generate a sequence of graphs from the given graph that capture a wide range of spectral and spatial behaviors.* We propose a new operator based on this derived sequence.

**Definition 2** (Variable power operator). Consider an unweighted and undirected graph $\mathcal{G}$. Let $\boldsymbol{A}^{[k]}$ denote the $k$-distance adjacency matrix, i.e., $\left[\boldsymbol{A}^{[k]}\right]_{ij} = 1$ if and only if the shortest distance (in the original graph) between nodes $i$ and $j$ is $k$. The variable power operator of order $r$ is defined as:

$$\boldsymbol{A}_{\boldsymbol{\theta}}^{(r)} = \sum_{k=0}^{r} \theta_k \boldsymbol{A}^{[k]}, \tag{3}$$

where $\boldsymbol{\theta} := (\theta_0, \ldots, \theta_r)$ is a set of parameters.

Clearly, $\boldsymbol{A}_{\boldsymbol{\theta}}^{(r)}$ is a natural extension of the classical adjacency matrix (i.e., $r = 1$ and $\theta_0 = \theta_1 = 1$). With power order $r > 1$, one can increase the spatial scope of information fusion on the graph when applying the convolution operation. The parameters $\theta_k$ also has a natural explanation—the magnitude and the sign of $\theta_k$ can be viewed as "global influence strength" and "global influence propensity" at distance $k$, respectively, which also determines the participation factor of each graph in the sequence in the aggregated operator.

Furthermore, we provide some theoretical justification of the proposed operator by establishing the following asymptotic property of spectral separation under the important SBM setting, which is, nevertheless, not enjoyed by the classical Laplacian operator or its normalized or powered versions. (All proofs are given in the appendix.)

**Theorem 3** (Asymptotic spectral separation of variable power operator). Consider a graph $\mathcal{G}$ drawn from $\text{SBM}(n, 2, a_{\text{intra}}, a_{\text{inter}})$. Assume that the signal-to-noise ratio $\xi_2^2/\xi_1 > 1$, where $\xi_1 = \frac{1}{2}(a_{\text{intra}} + a_{\text{inter}})$ and $\xi_2 = \frac{1}{2}(a_{\text{intra}} - a_{\text{inter}})$ (c.f., (Decelle et al., 2011)). Suppose $r$ is on the order of $c \log(n)$ for a constant $c$, such that $c \log(\xi_1) < 1/4$. Given nonvanishing $\theta_k$ for $k > r/2$, the variable power operator $\boldsymbol{A}_{\boldsymbol{\theta}}^{(r)}$ has the following spectral properties: **(i)** the leading eigenvalue is on the order of $\Theta\left(\|\boldsymbol{\theta}\|_1 \xi_1^r\right)$, the second leading eigenvalue is on the order of $\Theta\left(\|\boldsymbol{\theta}\|_1 \xi_2^r\right)$, and the rest are bounded by $\|\boldsymbol{\theta}\|_1 n^\epsilon \xi_1^{r/2} O(\log(n))$ for any fixed $\epsilon > 0$; and **(ii)** the two leading eigenvectors are sufficient to recover the two communities asymptotically (i.e., as $n$ goes to infinity).

Intuitively, the above theoretical result suggests that the variable power operator is able to *"magnify" benign signals* from the latent graph structure while *"suppressing" noises* due to random artefacts.

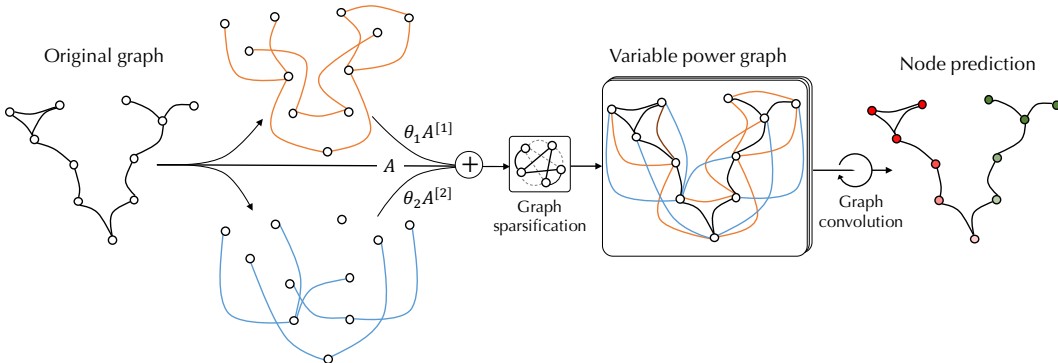

Figure 1: From the original graph, we generate a series of graphs, which are weighted by parameters that gauge the influence strengths, (optionally) sparsified, and eventually combined to form a variable power graph.

This is expected to improve spectral methods in general, especially when the benign signals tend to be overwhelmed by noises. For the rest of the paper, we will apply this insight to propose a robust graph learning paradigm in Section 2, as well as a new GCN architecture in Section 3. We also provide empirical evidence of the gain from this theory in Section 4 and conclude in Section 5.

### 1.3 RELATED WORK

**Beyond nearest neighbors.** Several works have been proposed to address the issue of limited spatial scope by powering the adjacency matrix (Lee et al., 2018; Wu et al., 2019; Li et al., 2019). However, simply powering the adjacency does not extract spectral gap and may even make the eigenspectrum more sensitive to perturbations. Abu-El-Haija et al. (2018; 2019) also introduced weight matrices for neighbors at different distances. But this could substantially increase the risk of overfitting in the low-data regime and make the network vulnerable to adversarial attacks.

**Robust spectral theory.** The robustness of spectral methods has been extensively studied for graph partitioning/clustering (Li et al., 2007; Balakrishnan et al., 2011; Chaudhuri et al.; Amini et al., 2013; Joseph et al., 2016; Diakonikolas et al., 2019). Most recently, operators based on self-avoiding or nonbacktracking walks have become popular for SBM (Massoulié, 2014; Mossel et al., 2013; Bordenave et al., 2015), which provably achieve the detection threshold conjectured by Decelle et al. (2011). Our work is partly motivated by the graph powering approach by Abbe et al. (2018), which leveraged the result of Massoulié (2014); Bordenave et al. (2015) to prove the spectral gap. The main difference with this line of work is that these operators are studied only for spectral clustering without incorporating nodal features. Our proposed variable power operator can be viewed as a kernel version of the graph powering operator (Abbe et al., 2018), thus substantially increasing the capability of learning complex nodal interactions while maintaining the spectral property.

**Robust graph neural network.** While there is a surge of adversarial attacks on graph neural networks (GNNs) (Dai et al., 2018; Zügner & Günnemann, 2019; Bojchevski & Günnemann, 2019), very few methods have been proposed for defense (Sun et al., 2018). Existing works employed known techniques from computer vision (Szegedy et al., 2014; Goodfellow et al., 2015; Szegedy et al., 2016), such as adversarial training with "soft labels" (Chen et al., 2019) or outlier detection in the hidden layers (Zhu et al., 2019), but they do not exploit the unique characteristics of graph-structured data. Importantly, our approach simultaneously improves performance in both the benign and adversarial tests, as shown in Figure 2 (details are presented in Section 4).

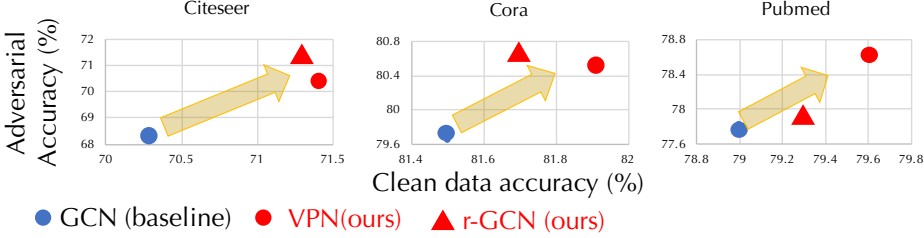

Figure 2: Our proposed framework can improve both clean and adversarial (10% attack by $\mathcal{A}_{DW_3}$ (Bojchevski & Günnemann, 2019)) accuracy for semi-supervised learning benchmarks.

## 2 GRAPH AUGMENTATION: ROBUST TRAINING PARADIGM

**Exploration of the spectrum of spectral and spatial behaviors.** Given a graph $\mathcal{G}$, consider a family of its powered graphs, $\{\mathcal{G}^{(2)}, \ldots, \mathcal{G}^{(r)}\}$, where $\mathcal{G}^{(k)}$ is obtained by connecting nodes with distance less than or equal to $k$. This "graph augmentation" procedure is similar to "data augmentation", because instead of limiting the learning on a single graph that is given, we artificially create a series of graphs that are closely related to each other in the spatial and spectral domains.

As we increase the power order, the graph also becomes more homogenized (Figure A.1). In particular, it can help near-isolated nodes (i.e., low-degree vertices), since they become connected beyond their nearest neighbors. By comparison, simply raising the adjacency matrix to its $r$-th power will

make them appear even more isolated, because it inadvertently promotes nodes with high degrees or nearby loops much more substantially as a result of feedback. Furthermore, the powered graphs can extract spectral gaps in the original graph despite local irregularities, thus boosting the signal-to-noise ratio in the spectral domain.

**Transfer of knowledge from the powered graph sequence.** Consider a generic learning task on a graph $\mathcal{G}$ with data $\mathcal{D}$. The loss function is denoted by $\ell(\mathcal{W}; \mathcal{G}, \mathcal{D})$ for a particular GCN architecture parametrized by $\mathcal{W}$. For instance, in semi-supervised learning, $\mathcal{D}$ consists of features and labels on a small set of nodes (see Table A.1), and $\ell$ is the cross-entropy loss over all labeled examples. Instead of minimizing over $\ell(\mathcal{W}; \mathcal{G}, \mathcal{D})$ alone, we use all the powered graphs:

$$\ell(\mathcal{W}; \mathcal{G}, \mathcal{D}) + \sum_{k=2}^{r} \alpha_k \ell(\mathcal{W}; \mathcal{G}^{(k)}, \mathcal{D}), \tag{r-GCN}$$

where $\alpha_k \geq 0$ gauges how much information one desires to transfer from powered graph $\mathcal{G}^{(k)}$ to the learning process. By minimizing the (r-GCN) objective, one seeks to optimize the network parameter $\mathcal{W}$ on multiple graphs simultaneously, which is beneficial in two ways: **(i)** in the *low-data regime*, like semi-supervised learning, it helps to reduce the variance to improve generalization and transferability; **(ii)** in the *adversarial setting*, it robustifies the network since it is more likely that the perturbed network is contained in the wider spectrum during training.

## 3 FROM FIXED TO VARIABLE POWER NETWORK

By using the variable power operator illustrated in Figure 1, we substantially increase the search space of graph operators. The proposed operator also leads to a new family of graph algorithms with broader scope of spatial fusion and enhanced spectral robustness. As the power grows, the network eventually becomes dense. To manage this increased complexity and make the network more robust against adversarial attacks in the feature domain, we propose a pruning mechanism.

**Graph sparsification.** Given a graph $\mathcal{G} := (\mathcal{V}, \mathcal{E}^{[1]})$, consider its powered version $\mathcal{G}^{(r)} := (\mathcal{V}, \mathcal{E}^{(r)})$ and a sequence of intermediate graphs $\mathcal{G}^{[2]}, \ldots, \mathcal{G}^{[r]}$, where $\mathcal{G}^{[k]} := (\mathcal{V}, \mathcal{E}^{[k]})$ is constructed by connecting two vertices if and only if the shortest distance is $k$ in $\mathcal{G}$. Clearly, $\{\mathcal{E}^{[k]}\}_{k=1}^{r}$ forms a partition of $\mathcal{E}^{(r)}$. For each node $i \in \mathcal{V}$, denote its $r$-neighborhood by $\mathcal{N}_r(i) := \{j \in \mathcal{V} \mid d_{\mathcal{G}}(i, j) \leq r\}$, which is identical to the set of nodes adjacent to $i$ in $\mathcal{G}^{[r]}$. Next, for each edge within this neighborhood, we associate a value using some suitable distance metric $\phi$ to measure "aloofness." For instance, it can be the usual Euclidean distance or correlation distance in the feature space. Based on this formulation, we prune an edge $e := (i, j)$ in $\mathcal{E}^{(r)}$ if the value is larger than a threshold $\tau(i, j)$, and denote the edge set after pruning $\bar{\mathcal{E}}^{(r)}$. Then, we can construct a new sequence of sparsified graphs, $\overline{\mathcal{G}}^{[k]}$ with edge sets $\bar{\mathcal{E}}^{[k]} = \mathcal{E}^{[k]} \cap \bar{\mathcal{E}}^{(r)}$ and adjacency matrix $\bar{A}^{[k]}$. Hence, the variable power operator is given by $\bar{A}_{\boldsymbol{\theta}}^{(r)} = \sum_{k=0}^{r} \theta_k \bar{A}^{[k]}$. Due to the influence of high-degree nodes in the spectral domain, one can *adaptively* choose the thresholds $\tau(i, j)$ to mitigate their effects. Specifically, we choose $\tau$ to be a small number if either $i$ or $j$ are nodes with high degrees, thereby making the sparsification more influential in densely connected neighborhoods than weakly connected parts.

**Layer-wise update rule.** To demonstrate the effectiveness of the proposed operator, we adopt the vanilla GCN strategy (2). Importantly, we replace the graph convolutional operator $\mathcal{A}$ with the variable power operator to obtain the variable power network (VPN):

$$\mathcal{A} = \boldsymbol{D}^{-\frac{1}{2}}(\boldsymbol{I} + \bar{\boldsymbol{A}}_{\boldsymbol{\theta}}^{(r)})\boldsymbol{D}^{-\frac{1}{2}}, \tag{VPN}$$

where $D_{ii} = 1 + |\{j \in \mathcal{V} \mid d_{\mathcal{G}}(i, j) = 1\}|$. The introduction of $\boldsymbol{I}$ is reminiscent of the "renormalization trick" (Kipf & Welling, 2017), but it can be also viewed as a regularization strategy in this context, which is well-known to improve the spectral robustness (Amini et al., 2013; Joseph et al., 2016). This construction immediately increases the scope of data fusion by a factor of $r$.

**Proposition 4.** By choosing $\mathcal{A}$ with (VPN) in the layer-wise update rule (2), the output at each node from a $L$-layer GCN depends on neighbors within $L * r$ hops.

Since we proved that the variable power operator has nice spectral separation in Theorem 3, VPN is expected to promote useful spectral signals from the graph topology (similar to the preservation of

useful information in images (Jacobsen et al., 2018), our method preserves useful information in the graph topology). This claim is substantiated with the following proposition.

**Proposition 5.** Given a graph with two hidden communities. Consider a 2-layer GCN architecture with layer-wise update rule (2). Suppose that $\mathcal{A}$ has a spectral gap. Further, assume that the leading two eigenvectors are asymptotically aligned with $\mathbf{1}$ and $\boldsymbol{\nu}$, i.e., the community membership vector, and that both are in the range of feature matrix $\boldsymbol{X}$. Then, there exists a configuration of $\boldsymbol{W}^{(1)}$ and $\boldsymbol{W}^{(2)}$ such that the GCN outputs can recover the community with high probability.

## 4 EXPERIMENTS

The proposed methods are evaluated against several recent models, including vanilla GCN (Kipf & Welling, 2017) and its variant PowerLaplacian where we simply replace the adjacent matrix with its powered version, three baselines using powered Laplacian IGCN (Li et al., 2019), SGC (Wu et al., 2019) and LNet (Liao et al., 2019), the recent method MixHop (Abu-El-Haija et al., 2019) which attempts to increase spatial scope fusion, as well as a state-of-the-art baseline and RGCN (Zhu et al., 2019), which is also aimed at improving the robustness of Vanilla GCN. All baseline methods on based on their public codes.

### 4.1 REVISITING THE STOCHASTIC BLOCK MODEL

**SBM dataset.** We generated a set of networks under SBM with 4000 nodes and parameters such that the SNRs range from 0.58 to 0.91. To disentangle the effects from nodal features with that from the spectral signal, we set the nodal features to be one-hot vectors. The label rates are 0.1%, 0.5% and 1%, the validation rate is 1%, and the rest of the nodes are testing points.

**Performance.** Since the nodal features do not contain any useful information, learning without topology such as multi-layer perceptron (MLP) is only as good as random guessing. The incorporation of graph topology improves classification performance—the higher the SNR (i.e., $\xi_2^2/\xi_1$, see Theorem 3), the higher the accuracy. Overall, as shown in Figure 3, the performance of the proposed method (VPN) is superior than other baselines, which either use Laplacian (GCN, Chebyshev, RGCN, LNet) or its powered variants (IGCN, PowerLaplacian).

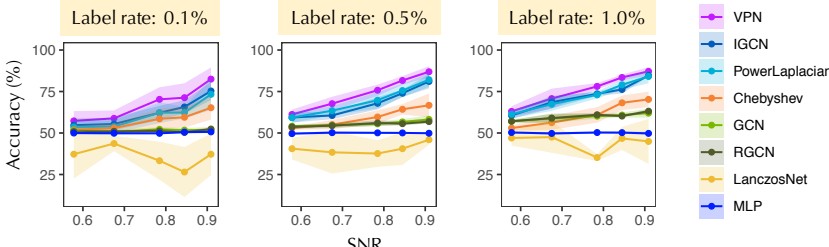

Figure 3: Comparison for SBM dataset. Shaded area indicates standard deviation over 10 runs.

**Spectral separation and Fourier modes.** From the eigenspectrum of the convolution operators (Figure A.3), we see that the spectral separation property is uniquely possessed by VPN, whose first two leading eigenvectors carry useful information about the underlying communities: without the help of nodal features, the accuracy is 87% even with label rate of 0.5%. Let $\boldsymbol{\Phi}$ denote the Fourier

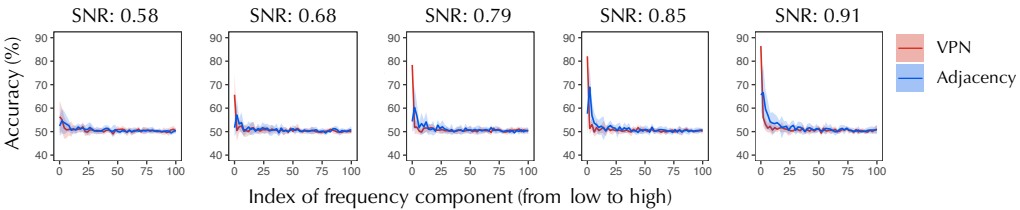

Figure 4: Accuracy of first 100 frequency components of VPN (red) and adjacency matrix (blue).

Table 1: Results for semi-supervised node classification. We highlighted the best and the second best performances, where we broke the tie by choosing the one with the smallest standard deviation.

| Model | Citeseer | Cora | Pubmed |
|---|---|---|---|
| ManiReg (Belkin et al., 2006) | 60.1 | 59.5 | 70.7 |
| SemiEmb (Weston et al., 2012) | 59.6 | 59.0 | 71.1 |
| LP (Zhu et al., 2003) | 45.3 | 68.0 | 63.0 |
| DeepWalk (Perozzi et al., 2014) | 43.2 | 67.2 | 65.3 |
| ICA (Lu & Getoor, 2003) | 69.1 | 75.1 | 73.9 |
| Planetoid (Yang et al., 2016) | 64.7 | 75.7 | 77.2 |
| Vanilla GCN (Kipf & Welling, 2017) | 70.3 | 81.5 | 79.0 |
| PowerLaplacian | 70.5 | 80.5 | 78.3 |
| IGCN(RNM) (Li et al., 2019) | 69.0 | 80.9 | 77.3 |
| IGCN(AR) (Li et al., 2019) | 69.3 | 81.1 | 78.2 |
| LNet (Liao et al., 2019) | $66.2 \pm 1.9$ | $79.5 \pm 1.8$ | $78.3 \pm 0.3$ |
| RGCN (Zhu et al., 2019) | $71.2 \pm 0.5$ | $\mathbf{82.8} \pm 0.6$ | $79.1 \pm 0.3$ |
| SGC (Wu et al., 2019) | $\mathbf{71.9} \pm 0.1$ | $81.0 \pm 0.0$ | $78.9 \pm 0.0$ |
| MixHop (Abu-El-Haija et al., 2019) | $71.4 \pm 0.8$ | $81.9 \pm 0.4$ | $\mathbf{80.8} \pm 0.6$ |
| r-GCN (this paper) | $71.3 \pm 0.54$ | $81.7 \pm 0.23$ | $79.3 \pm 0.31$ |
| VPN (this paper) | $\mathbf{71.4} \pm 0.57$ | $\mathbf{81.9} \pm 0.32$ | $\mathbf{79.6} \pm 0.39$ |

modes of the the adjacency matrix or VPN, and $X$ be the nodal features (i.e., identity matrix). We analyze the information from spectral signals (e.g., the $k$-th and $k+1$-th eigenvectors) by estimating the accuracy of an MLP with filtered nodal features, namely $\mathbf{\Phi}_{:,k:(k+1)} \mathbf{\Phi}_{:,k:(k+1)}^{\top} X$, as shown in Figure 4. The accuracy reflects the information content in the frequency components. We see that the two leading eigenvectors of VPN are sufficient to perform classification, whereas those of the adjacency matrix cannot make accurate inferences.

## 4.2 SEMI-SUPERVISED NODE CLASSIFICATION

**Experimental setup.** We followed the setup of (Yang et al., 2016; Kipf & Welling, 2017) for citation networks Citeseer, Cora and Pubmed (please refer to the Appendix for more details).

**Graph powering order** can influence spatial and spectral behaviors. Our theory suggests powering to the order of $\log(n)$; in practice, orders of 2 to 4 suffice (Figure 5). Here, we chose the power order to be 4 for r-GCN on Citeseer and Cora, and 3 for Pubmed, and reduced the order by 1 for VPN.

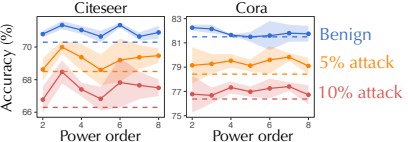

Figure 5: Benign and adversarial accuracy as power order increases. Dahsed lines correspond to vanilla GCN.

**Performance.** By replacing Laplacian with VPN, we see an immediate improvement in performance (Table 1). We also see that a succinct parametrization of the global influence relation in VPN is able to increase the expressivity of the network. For instance, the learned $\theta$ at distances 2 and 3 for Citeseer are 3.15e-3 and 3.11e-3 with $p$-value less than 1e-5. This implies that the network tends to put more weights in closer neighbors.

## 4.3 DEFENSE AGAINST EVASION ATTACKS

To evaluate the robustness of the learned network, we considered the setting of evasion attacks, where the model is trained on benign data but tested on adversarial data.

**Adversarial methods.** Five strong global attack methods are considered, including DICE (Zügner & Günnemann, 2019), $\mathcal{A}_{abr}$ and $\mathcal{A}_{DW_3}$ (Bojchevski & Günnemann, 2019), Meta-Train and Meta-Self (Zügner & Günnemann, 2019). We further modulated the severity of attack methods by varying the attack rate, which corresponds to the percentage of edges to be attacked.

**Robustness evaluation.** In general, both r-GCN and VPN are able to improve over baselines for the defense against evasion attacks, e.g., Figure 6 for the $\mathcal{A}_{DW_3}$ attack (detailed results for other attacks are listed in the Appendix). It can be also observed that the proposed methods are more robust in Citeseer and Cora than Pubmed. In addition to the low label rates, we conjectured that topological

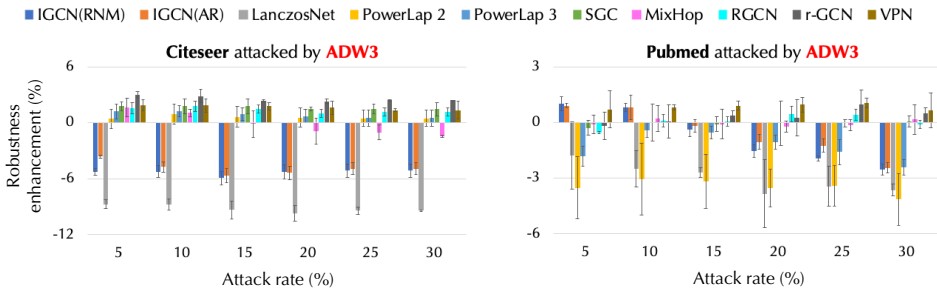

Figure 6: Robustness merit $\left( \text{accuracy}_{\text{proposed-method}}^{\text{post-attack}} - \text{accuracy}_{\text{vanilla GCN}}^{\text{post-attack}} \right)$ reported in percentage under $\mathcal{A}_{DW_3}$ attack. The error bar indicates standard deviation over 20 independent simulations.

attacks are more difficult to defend for networks with prevalent high-degree nodes, because the attacker can bring in more irrelevant vertices by simply adding a link to the high-degree nodes.

**Informative and robust low-frequency spectral signal.** It has been observed by Wu et al. (2019); Maehara (2019) that GCNs share similar characteristics of low-pass filters, in the sense that nodal features filtered by low-frequency Fourier modes lead to accurate classifiers (e.g., MLP). However, one key question left unanswered is how to obtain the Fourier modes. In their experiments, they derive it from the graph Laplacian. By using VPN to construct the Fourier modes, we show that the information content in the low-frequency domain can be improved.

More specifically, we first perform eigendecomposition of the graph convolutional operator (i.e., graph Laplacian or VPN) to obtain the Fourier modes $\mathbf{\Phi}$. We then reconstruct the nodal features $\boldsymbol{X}$ using only the $k$-th and the $k + 1$-th eigenvectors, i.e., $\mathbf{\Phi}_{:,k:(k+1)} \mathbf{\Phi}_{:,k:(k+1)}^{\top} \boldsymbol{X}$. We then use the reconstructed features in MLP to perform the classification task in a supervised learning setting. As Figure 7 shows, features filtered by the leading eigenvectors of VPN lead to higher classification accuracy compared to the classical adjacency matrix.

For the adversarial testing, we construct a new basis $\widetilde{\mathbf{\Phi}}$ based on the attacked graph, and then use $\widetilde{\mathbf{\Phi}}_{:,k:(k+1)} \widetilde{\mathbf{\Phi}}_{:,k:(k+1)}^{\top} \boldsymbol{X}$ as test points for the MLP trained in the clean data setting. As can be seen in Figure 8, models trained based on VPN filtered features also have better adversarial robustness in evasion attacks. Since the eigenvalues of the corresponding operator exhibit low-pass filtering characteristics (Figure A.2), the enhanced benign and adversarial accuracy of VPN is attributed to the increased signal-to-noise ratio in the low-frequency domain. This is in alignment with the theoretical proof of spectral gap developed in this study.

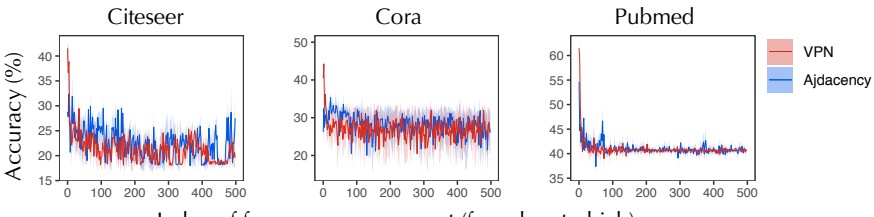

Figure 7: Accuracy of first 500 frequency components for VPN (red) and adjacency matrix (blue). The x axis corresponds to the index $k$ in $\mathbf{\Phi}_{:,k:(k+1)} \mathbf{\Phi}_{:,k:(k+1)}^{\top} \boldsymbol{X}$ for signal reconstruction.

## 4.4 DISCUSSIONS

In this paper, we proposed a new graph operator to replace the graph Laplacian in the classic GCN. It is worth mentioning that our approach is very different from the $k$-th order polynomials of the Laplacian developed in (Defferrard et al., 2016) or the high-order graph Laplacian employed in (Lee et al., 2018; Li et al., 2019; Abu-El-Haija et al., 2019; Wu et al., 2019). If the graph Laplacian $\boldsymbol{L}$ is powered to the $k$-th order, the resulting matrix has the same eigenvectors as $\boldsymbol{L}$, with only the eigenvalues

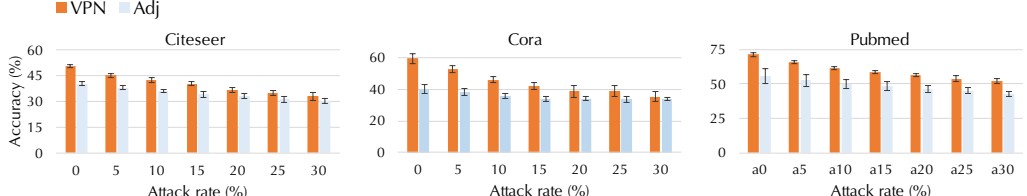

Figure 8: Accuracy of the leading frequency components in adversarial testing. We use $\widetilde{\mathbf{\Phi}}_{:,1:k}\widetilde{\mathbf{\Phi}}_{:,1:k}^{\top}\boldsymbol{X}$ as reconstructed nodal features for testing, where $k$ is 10 for Citeseer and 5 for Cora and Pubmed. Error bar indicates standard deviation over 10 runs.

powered to the $k$-th order. Thus, the corresponding $k$-th order polynomial has the same eigenvectors, with the polynomial function applied only to the eigenvalues. However, the proposed variable power operator has radically different eigenvectors as the graph Laplacian or its powered variant. This is an important difference because we observed empirically that the leading eigenvectors of a graph Laplacian are extremely sensitive to outliers under the SBM, and they often correspond to either tails or high-degree nodes (please see Figure A.3 for an illustration). However, our result in Theorem 3 suggests that the leading eigenvectors of the proposed operator asymptotically recovers the underlying community under SBM, and enjoys the "spectral gap" property.

We also remark that the meaning of robustness in the machine learning literature ("adversarial robustness") and in the spectral theory literature ("spectral robustness") appear to be different. However, in this work, we argue that spectral robustness is closely related to adversarial robustness. This can be explained from a matrix perturbation point of view. Let $\lambda_k$ and $\boldsymbol{\phi}_k$ be the eigenvalue and the corresponding eigenvector of the adjacency matrix $\boldsymbol{A}$. Then, the sensitivity of the eigenvector to the perturbation is given by (Trefethen & Bau III, 1997):

$$\frac{\partial \boldsymbol{\phi}_k}{\partial \boldsymbol{A}_{(ij)}} = \sum_{\ell=1, \ell \neq k}^{n} \frac{\phi_{\ell(i)}\phi_{k(j)}(2 - \delta_{(ij)})}{\lambda_k - \lambda_\ell}\boldsymbol{\phi}_\ell,$$

where $\boldsymbol{A}_{(ij)}$ is the $(i, j)$-th entry of $\boldsymbol{A}$, $\phi_{k(j)}$ is the $j$-th entry of $\boldsymbol{\phi}_k$, $\delta_{(ij)}$ is the perturbation of the $(i, j)$-th entry of $\boldsymbol{A}$, and $n$ is the dimension of $\boldsymbol{A}$. It can be seen that the perturbation of the convolutional operator is controlled by the inverse of the spectral gap. When the spectral gaps between the leading eigenvalues (i.e., $\lambda_1$ and $\lambda_2$) and the rest of the eigenvalues are large, the perturbations of the leading eigenvectors due to the adversarial attack are controlled. Since the eigenvalues of the graph convolutional operator decays very quickly (see Figure A.2 as an illustration), the learned representation is heavily influenced by the leading eigenvectors. Thus, by controlling the perturbations of the leading eigenvectors, one can expect to control the perturbations of the outputs.

Our theoretical result in Theorem 3 was based on the standard SBM, which is a classic model proposed in mathematical sociology and has been adopted to model and analyze real-world social and biological networks (Funke & Becker, 2019). The main advantage of SBM is that it encodes the structural information that nodes belong to the same community tend to be more connected with each other. Nevertheless, we recognize one limitation of our analysis is that SBM might not be a suitable model for some existing or future applications, in which case, models with more flexibility in the degree distributions, such as the degree-corrected SBM proposed by Karrer & Newman (2011), labeled SBM by Heimlicher et al. (2012), or hiearchical SBM proposed by Peixoto (2017) could be more applicable. Also our analysis is limited to two communities, and the extension to more than two communities is a challenging open problem. Nevertheless, through extensive experiments, we observed that our theory has strong implications for real-world graphs. This is evident from the informative and robust low-frequency spectral signal results in Figures 7 and 8, which show that the proposed graph convolution operator has a clear advantage in the low-frequency regime over the classic graph Laplacian in terms of classification accuracy.

## 5 CONCLUSION

This study goes beyond classical spectral graph theory to defend GCNs against adversarial attacks. We challenge the central building block of existing methods, namely the graph Laplacian, which

is not robust to noisy links. For adversarial robustness, spectral separation is a desirable property. We propose a new operator that enjoys this property and can be incorporated in GCNs to improve expressivity and interpretability. Furthermore, by generating a sequence of powered graphs based on the original graph, we can explore a spectrum of spectral and spatial behaviors and encourage transferability across graphs. The proposed methods are shown to improve both benign and adversarial accuracy over various baselines evaluated against a comprehensive set of attack strategies.

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

## A  APPENDIX

### A.1  PROOF OF PROPOSITION 4

To validate the claim, we prove a more general result. Consider a graph $\mathcal{G}$ and its powered graph $\mathcal{G}^{(r)}$, whose adjacency matrices are $\boldsymbol{A}$ and $\boldsymbol{A}^{(r)}$, respectively. Since the support of $\boldsymbol{A}_{\boldsymbol{\theta}}^{(r)}$ is identical to that of $\boldsymbol{A}^{(r)}$, we will focus on $\boldsymbol{A}^{(r)}$ for notational simplicity. For any signal $\boldsymbol{f} : \mathcal{V} \mapsto \mathbb{R}^{|\mathcal{V}|}$ on a graph, denote the support of the signal $f_v$ on node $v \in \mathcal{V}$ by $\mathcal{S}_v^f$, i.e., $f_v$ only depends on the signals on nodes in $\mathcal{S}_v^f$. Then, it is easy to see that the signal $\boldsymbol{g} \triangleq \boldsymbol{A}\boldsymbol{f}$ has support $\mathcal{S}_v^g \subseteq \cup_{u \sim_r v} \mathcal{S}_u^f$, where $u \sim_r v$ denotes adjacency relation on $\mathcal{G}^{(r)}$ defined by $\boldsymbol{A}^{(r)}$. Clearly, the set defined by $\{u \in \mathcal{V} \mid u \sim_r v\}$ is identical to $\{u \in \mathcal{V} \mid d_{\mathcal{G}}(u,v) \leq r\}$, where $d_{\mathcal{G}}(u,v)$ denotes the shortest distance from $u$ to $v$ on $\mathcal{G}$.

Also, we see that element-wise operation does not expand the support, i.e., for any element-wise activation $\sigma$, the signal $\boldsymbol{g}_\sigma \triangleq \sigma(\boldsymbol{f})$ has support $\mathcal{S}_v^{g_\sigma} \subseteq \mathcal{S}_v^f$. In particular, we can prove the claim by iteratively applying the function composition layer by layer, since each layer is of the form $\sigma(\boldsymbol{A}_{\boldsymbol{\theta}}^{(r)} \boldsymbol{H}^{(l)} \boldsymbol{W}^{(l)})$, where $\boldsymbol{H}^{(l)}$ is the hidden state at layer $l$ and $\boldsymbol{W}^{(l)}$ is the corresponding weight matrix. Hence after $L$ iterations of layer-wise updates, the support of the output on $v$ is given by $\{u \in \mathcal{V} \mid d_{\mathcal{G}}(u,v) \leq r * L\}$.

### A.2  PROOF OF PROPOSITION 5

Let $\lambda_1$ and $\lambda_2$ be the two leading eigenvalues of $\bar{\boldsymbol{A}}$ with corresponding eigenvectors $\phi_1$ and $\phi_2$, respectively. Without loss of generality, assume that both eigenvalues are nonnegative. Under the

assumption that $\phi_1$ and $\phi_2$ lie in the range of $X$, there exists a $W^{(1)}$ such that $XW^{(1)} = [\phi_1 \quad \phi_2]$. So the output of the first layer with ReLU activation is given by:

$$H^{(1)} = \sigma\left(\bar{A}XW^{(1)}\right) = \sigma\left(\bar{A}[\phi_1 \quad \phi_2]\right) \tag{4}$$

$$= [\lambda_1\sigma(\phi_1) \quad \lambda_2\sigma(\phi_2)] \tag{5}$$

$$= \left[\lambda_1(\mathbf{1} + \sigma(h_1)) \quad \lambda_2(\tfrac{1}{2}\nu + \tfrac{1}{2}\mathbf{1} + \sigma(h_2))\right] \quad \text{where } \|h_1\| = o(1) \text{ and } \|h_2\| = o(1) \tag{6}$$

where (5) is due to the definition of eigenvectors and that $\sigma$ is an element-wise operation, and (6) is due to the asymptotic alignment condition and the fact that $\sigma(\nu) = \frac{1}{2}\nu + \frac{1}{2}\mathbf{1}$. Therefore, by choosing the last layer weight $W^{(2)} = \begin{bmatrix} -\frac{1}{\lambda_1^2\lambda_2} & \frac{1}{\lambda_1\lambda_2^2} \\ \frac{1}{\lambda_2^2} & -\frac{2}{\lambda_2^2} \end{bmatrix}$, the output of the last layer is given by:

$$Y = \bar{A}H^{(1)}W^{(2)}$$

$$= \bar{A}\left[\lambda_1(\mathbf{1} + \sigma(h_1)) \quad \lambda_2(\tfrac{1}{2}\nu + \tfrac{1}{2}\mathbf{1} + \sigma(h_2))\right]W^{(2)}$$

$$= \left[\lambda_1^2\mathbf{1} + h_1' \quad \tfrac{\lambda_2^2}{2}\nu + \tfrac{\lambda_1\lambda_2}{2}\mathbf{1} + h_2'\right]W^{(2)}$$

$$= [\nu + h_1'' \quad -\nu + h_2'']$$

where $\|h_k'\|_2 = o(1), \|h_k''\|_2 = o(1)$ for $k = 1, 2$. Therefore, after the softmax operation, we can see that the 2-layer output recovers the membership of the elements exactly.

One general remark is that while the above argument is not limited to the leading two eigenvalues/eigenvectors, from the form of $W^{(2)}$ whose elements depend inversely on the corresponding eigenvalues, we can expect more numerical stability (and hence easier learning) when the corresponding eigenvalues are large.

### A.3 PROOF OF THEOREM 3

Consider a stochastic block model with two communities, where the intra-community connection probability is $a_{\text{intra}}/n$ and the inter-community connection probability is $a_{\text{inter}}/n$, and assume that $a_{\text{intra}} > a_{\text{inter}}$. Let $\xi_1 = \frac{a_{\text{intra}}+a_{\text{inter}}}{2}$ and $\xi_2 = \frac{a_{\text{intra}}-a_{\text{inter}}}{2}$, which correspond to the first and second eigenvalues of the expected adjacency matrix of the above model. Let $\nu$ denote the community label vector, with $\nu_i = \pm 1$ depending on the membership of node $i$. For weak recovery, we assume the detectability condition (Decelle et al., 2011):

$$\xi_2^2 > \xi_1. \tag{7}$$

With the notations from the main paper, let $A_\theta^{(r)} = \sum_{k=0}^{r} \theta_k A^{[k]}$ denote the $\theta$-weighted variable power operator, where $\left[A^{[k]}\right]_{ij} = \begin{cases} 1 & \text{if } d_\mathcal{G}(i,j) = k \\ 0 & \text{o.w.} \end{cases}$ is the $k$-distance adjacency matrix. Let $A^{\{k\}}$ denote the nonbacktracking path counting matrix, where the $(i, j)$-th component indicate the number of self-avoiding paths of graph edges of length $k$ connecting $i$ to $j$ (Massoulié, 2014). Our goal is to prove that the top eigenvectors of $A_\theta^{(r)}$ are asymptotically aligned with those of $A^{\{k\}}$ for $k$ greater than $\log(n)$ up to a constant multiplicative factor. To do so, we first recall the result of (Massoulié, 2014), which examines the behaviors of top eigenvectors of $A^{\{k\}}$.

**Lemma 6** ((Massoulié, 2014)). *Assume that (7) holds. Then, w.h.p., for all $k \in \{r/2, ..., r\}$:*

(a) *The operator norm $\|A^{\{k\}}\|_2$ is up to logarithmic factors $\Theta(\xi_1^k)$. The second eigenvalue of $A^{\{k\}}$ is up to logarithmic factors $\Omega(\xi_2^k)$.*

(b) *The leading eigenvector is asymptotically aligned with $A^{\{k\}}\mathbf{1}$:*

$$\frac{A^{\{k\}}A^{\{k\}}\mathbf{1}}{\|A^{\{k\}}A^{\{k\}}\mathbf{1}\|_2} = \frac{A^{\{k\}}\mathbf{1}}{\|A^{\{k\}}\mathbf{1}\|_2} + h_k$$

*where $\|h_k\| = o(1)$. The second eigenvector is asymptotically aligned with $A^{\{k\}}\nu$:*

$$\frac{A^{\{k\}}A^{\{k\}}\nu}{\|A^{\{k\}}A^{\{k\}}\nu\|_2} = \frac{A^{\{k\}}\nu}{\|A^{\{k\}}\nu\|_2} + h_k'$$

*where $\|h_k'\| = o(1)$.*

Now, we first define $\boldsymbol{\Gamma}_{\boldsymbol{\theta}}^{\{r\}} = \sum_{k=r/2}^{r} \theta_k \boldsymbol{A}^{\{k\}}$, so

$$\boldsymbol{A}_{\boldsymbol{\theta}}^{(r)} = \boldsymbol{\Gamma}_{\boldsymbol{\theta}}^{\{r\}} + \boldsymbol{\Delta}_{\boldsymbol{\theta}}^{\{r\}} = \boldsymbol{\Gamma}_{\boldsymbol{\theta}}^{\{r\}} + \boldsymbol{A}_{\boldsymbol{\theta}}^{(r/2-1)} + \sum_{k=r/2}^{r} \theta_k (\boldsymbol{A}^{[k]} - \boldsymbol{A}^{\{k\}}).$$

Also, by (Massoulié, 2014, Theorem 2.3), the local neighborhoods of vertices do not grow fast, and w.h.p., the graph of $\boldsymbol{A}^{(r/2-1)}$ has maximum degree $O(\xi_1^{r/2}(\log n)^2)$. Let $\phi$ be the leading eigenvector of $\boldsymbol{A}_{\boldsymbol{\theta}}^{(r/2-1)}$, and let $v$ be the node where $\phi$ is maximum (i.e., $\phi(v) \geq \phi(u)$). Without loss of generality, assume that $\phi(v) > 0$. We have a good control over the first eigenvalue $\lambda_1$ of $\boldsymbol{A}_{\boldsymbol{\theta}}^{(r/2-1)}$ as follows:

$$\lambda_1 = \frac{(\boldsymbol{A}_{\boldsymbol{\theta}}^{(r/2-1)}\phi)(v)}{\phi(v)} = \frac{\sum_{k=0}^{r/2-1} \sum_{\{u:d_{\mathcal{G}}(u,v)=k\}} \theta_k \phi(u)}{\phi(v)}$$

$$= \sum_{k=0}^{r/2-1} \theta_k \sum_{\{u:d_{\mathcal{G}}(u,v)=k\}} \frac{\phi(u)}{\phi(v)}$$

$$\leq \sum_{k=0}^{r/2-1} |\theta_k| \sum_{\{u:d_{\mathcal{G}}(u,v)=k\}} 1$$

$$\leq \|\boldsymbol{\theta}\|_\infty \sum_{\{u:d_{\mathcal{G}}(u,v)\leq r/2-1\}} 1$$

$$= O(\|\boldsymbol{\theta}\|_1 \xi_1^{r/2}(\log n)^2)$$

Due to (Abbe et al., 2018, Theorem 2), which proves an identical result of Lemma 6 but for $\boldsymbol{A}^{[k]}$, we know that w.h.p., for $r = \epsilon \log(n)$, we can bound the last term by

$$\| \sum_{k=r/2}^{r} \theta_k (\boldsymbol{A}^{[k]} - \boldsymbol{A}^{\{k\}}) \|_2 \leq \sum_{k=0}^{r} |\theta_k| \|\boldsymbol{A}^{[k]} - \boldsymbol{A}^{\{k\}}\|_2 = O\left( \|\boldsymbol{\theta}\|_1 \xi_1^{r/2}(\log n)^4 \right)$$

Therefore, by triangle inequality, we can conclude that

$$\|\boldsymbol{\Delta}_{\boldsymbol{\theta}}^{\{r\}}\|_2 = \|\boldsymbol{A}_{\boldsymbol{\theta}}^{(r)} - \boldsymbol{\Gamma}_{\boldsymbol{\theta}}^{\{r\}}\|_2 = O\left( \|\boldsymbol{\theta}\|_1 \xi_1^{r/2}(\log n)^4 \right). \tag{8}$$

Now, our goal is to show a similar result as Lemma 6 for $\boldsymbol{\Gamma}_{\boldsymbol{\theta}}^{\{r\}}$. In particular,

$$\frac{\|\boldsymbol{\Gamma}_{\boldsymbol{\theta}}^{\{r\}}\boldsymbol{A}^{\{r\}}\mathbf{1}\|_2}{\|\boldsymbol{A}^{\{r\}}\mathbf{1}\|_2} = \left\| \sum_{k=r/2}^{r} \theta_k \boldsymbol{A}^{\{k\}} \frac{\boldsymbol{A}^{\{r\}}\mathbf{1}}{\|\boldsymbol{A}^{\{r\}}\mathbf{1}\|_2} \right\|_2 \tag{9}$$

$$= \left\| \sum_{k=r/2}^{r} \theta_k \boldsymbol{A}^{\{k\}} \frac{\boldsymbol{A}^{\{k\}}\mathbf{1}}{\|\boldsymbol{A}^{\{k\}}\mathbf{1}\|_2} \right\|_2 + O\left( \sum_{k=r/2}^{r} |\theta_k| \|\boldsymbol{A}^{\{k\}}\|_2 \left\| \frac{\boldsymbol{A}^{\{r\}}\mathbf{1}}{\|\boldsymbol{A}^{\{r\}}\mathbf{1}\|_2} - \frac{\boldsymbol{A}^{\{k\}}\mathbf{1}}{\|\boldsymbol{A}^{\{k\}}\mathbf{1}\|_2} \right\| \right) \tag{10}$$

$$= \left\| \sum_{k=r/2}^{r} \theta_k \boldsymbol{A}^{\{k\}} \frac{\boldsymbol{A}^{\{k\}}\mathbf{1}}{\|\boldsymbol{A}^{\{k\}}\mathbf{1}\|_2} \right\|_2 + O\left( \sum_{k=r/2}^{r} |\theta_k| \xi_1^k n^{-\delta} \right) \tag{11}$$

$$= \left\| \sum_{k=r/2}^{r} \theta_k \boldsymbol{A}^{\{k\}} \frac{\boldsymbol{A}^{\{k\}}\mathbf{1}}{\|\boldsymbol{A}^{\{k\}}\mathbf{1}\|_2} \right\|_2 + O\left( \|\boldsymbol{\theta}\|_1 \xi_1^r n^{-\delta} \right) \tag{12}$$

$$= \left\| \sum_{k=r/2}^{r} \theta_k \left( \frac{\boldsymbol{A}^{\{k\}}\mathbf{1}}{\|\boldsymbol{A}^{\{k\}}\mathbf{1}\|_2} + \boldsymbol{h}_k \right) \frac{\|\boldsymbol{A}^{\{k\}}\boldsymbol{A}^{\{k\}}\mathbf{1}\|_2}{\|\boldsymbol{A}^{\{k\}}\mathbf{1}\|_2} \right\|_2 + O\left( \|\boldsymbol{\theta}\|_1 \xi_1^r n^{-\delta} \right) \quad \text{for } \|\boldsymbol{h}_k\|_2 = o(1)$$

$$\tag{13}$$

$$= \left\| \sum_{k=r/2}^{r} \theta_k \left( \frac{A^{\{r\}}1}{\|A^{\{r\}}1\|_2} + h_k \right) \frac{\|A^{\{k\}}A^{\{k\}}1\|_2}{\|A^{\{k\}}1\|_2} \right\|_2 +$$
$$O \left( \left\| \sum_{k=r/2}^{r} \theta_k \frac{\|A^{\{k\}}A^{\{k\}}1\|_2}{\|A^{\{k\}}1\|_2} \right\|_2 \left\| \frac{A^{\{r\}}1}{\|A^{\{r\}}1\|_2} - \frac{A^{\{k\}}1}{\|A^{\{k\}}1\|_2} \right\| \right) + O \left( \|\theta\|_1 \xi_1^r n^{-\delta} \right)$$

$$\hspace{14cm} (14)$$

$$= \left\| \sum_{k=r/2}^{r} \theta_k \left( \frac{A^{\{r\}}1}{\|A^{\{r\}}1\|_2} + h_k \right) \frac{\|A^{\{k\}}A^{\{k\}}1\|_2}{\|A^{\{k\}}1\|_2} \right\|_2 + O \left( \|\theta\|_1 \xi_1^r n^{-\delta} \right) \tag{15}$$

$$= \left( \sum_{k=r/2}^{r} |\theta_k| \frac{\|A^{\{k\}}A^{\{k\}}1\|_2}{\|A^{\{k\}}1\|_2} \right) (1 + o(1)) + O \left( \|\theta\|_1 \xi_1^r n^{-\delta} \right) \tag{16}$$

$$= O \left( \|\theta\|_1 \xi_1^r n^{-\delta} \right), \tag{17}$$

where (10) and (14) are due to triangle inequalities, (11) and (15) are due to results in (Massoulié, 2014), and (13) is due to Lemma 6. Therefore, we have $\|\Gamma_\theta^{\{r\}} A^{\{r\}}1\|_2 = \Theta \left( \|\theta\|_1 \xi_1^r \|A^{\{r\}}1\|_2 \right)$. Similarly, we can prove that $\|\Gamma_\theta^{\{r\}} A^{\{r\}}\nu\|_2 = \Theta \left( \|\theta\|_1 \xi_2^r \|A^{\{r\}}\nu\|_2 \right)$.

Furthermore, we can show that $\Gamma_\theta^{\{r\}}$ satisfies the weak Ramanujan property (the proof is similar to (Massoulié, 2014) and is deferred to Section A.4).

**Lemma 7.** For any fixed $\epsilon > 0$, $\Gamma_\theta^{\{r\}}$ satisfies the following weak Ramanujan property w.h.p.,

$$\sup_{\|u\|_2=1, u^\top A^{\{r\}}1 = u^\top A^{\{r\}}\nu = 0} \|\Gamma_\theta^{\{r\}} u\|_2 = \|\theta\|_1 n^\epsilon \xi_1^{r/2} O(\log(n))$$

By Lemma 7, the leading two eigenvectors of $\Gamma_\theta^{\{r\}}$ will be asymptotically in the span of $A^{\{r\}}1$ and $A^{\{r\}}\nu$ according to the variational definition of eigenvalues. By our previous analysis, this means that the top eigenvalue of $\Gamma_\theta^{\{r\}}$ will be $\Theta (\|\theta\|_1 \xi_1^r)$ with eigenvector asymptotically aligned with $A^{\{r\}}1$. Since by (Massoulié, 2014, Lemma 4.4), $A^{\{r\}}1$ and $A^{\{r\}}\nu$ are asymptotically orthogonal, the second eigenvalue of $\Gamma_\theta^{\{r\}}$ will be $\Theta (\|\theta\|_1 \xi_2^r)$ with eigenvector asymptotically aligned with $A^{\{r\}}\nu$.

Since the perturbation term $\|\Delta_\theta^{\{r\}}\|_2 = o \left( \lambda_2(\Gamma_\theta^{\{r\}}) \right)$ by (8), using Weyl's inequality (Weyl, 1912), we can conclude that the leading eigenvalue $\lambda_1 \left( A_\theta^{(r)} \right) = \Theta (\|\theta\|_1 \xi_1^r)$, the second eigenvalue $\lambda_2 \left( A_\theta^{(r)} \right) = \Theta (\|\theta\|_1 \xi_2^r)$, and the rest of the eigenvalues are bounded by $\|\theta\|_1 n^\epsilon \xi_1^{r/2} O(\log(n))$. Moreover, since $\|\Delta_\theta^{\{r\}}\|_2 = o \left( \max \left\{ \lambda_1(\Gamma_\theta^{\{r\}}) - \lambda_2(\Gamma_\theta^{\{r\}}), \lambda_2(\Gamma_\theta^{\{r\}}) - \lambda_3(\Gamma_\theta^{\{r\}}) \right\} \right)$, by the Davis-Kahan Theorem (Davis & Kahan, 1970), the leading two eigenvectors of $A_\theta^{(r)}$ are asymptotically aligned with $A^{\{r\}}1$ and $A^{\{r\}}\nu$, which is shown to be enough for the rounding procedure of (Massoulié, 2014) to achieve weak recovery.

## A.4 PROOF OF LEMMA 7

Denote $\widetilde{A} = \frac{a_{\text{intra}}}{n} \left[ \frac{1}{2}(11^\top + \nu\nu^\top) - I \right] + \frac{a_{\text{inter}}}{2n}(11^\top - \nu\nu^\top)$ as the expected adjacency matrix conditioned on community labels $\nu$. Let $P_{ij}$ denote the set of all self-avoiding paths $i_0^k := \{i_0, \ldots, i_k\}$ from $i$ to $j$, such that no nodes appear twice in the path, and define $Q_{ij}^{\{m\}} := \sum_{i_0^m \in P_{ij}} \prod_{t=1}^{m}(A - \widetilde{A})_{i_{t-1}i_t}$. Let $T_{ij}^m$ be the concatenation of self-avoiding paths $i_0^{k-m}$ and $i_{k-m+1}^k$, and let $R_{ij}^m$ denote $T_{ij}^m \setminus P_{ij}$. Define $W_{ij}^{\{k,m\}} := \sum_{i_0^k \in R_{ij}^m} \prod_{t=1}^{k-m}(A - \widetilde{A})_{i_{t-1}i_t} \widetilde{A}_{i_{k-m}i_{k-m+1}} \prod_{t=k-m+2}^{k}(A)_{i_{t-1}i_t}$.

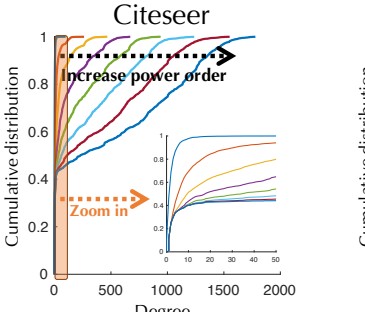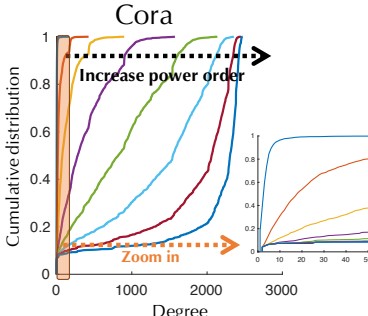

Figure A.1: Degree distribution in the original graph and the powered graphs for Citeseer and Cora. The graph becomes more homogeneous and high degree nodes become more prevalent as we increase the powers.

Then, by (Massoulié, 2014, Theorem 2.2), we can have

$$\boldsymbol{A}^{\{k\}} = \boldsymbol{Q}^{\{k\}} + \sum_{m=1}^{k}(\boldsymbol{Q}^{\{k-m\}}\widetilde{\boldsymbol{A}}\boldsymbol{A}^{\{m-1\}}) - \sum_{m=1}^{k}\boldsymbol{W}^{\{k,m\}}.$$

Hence,

$$\boldsymbol{\Gamma}_{\boldsymbol{\theta}}^{\{r\}} = \sum_{k=r/2}^{r} \theta_k \boldsymbol{A}^{\{k\}}$$

$$= \sum_{k=r/2}^{r} \theta_k \boldsymbol{Q}^{\{k\}} + \sum_{k=r/2}^{r} \theta_k \sum_{m=1}^{k} \left(\boldsymbol{Q}^{\{k-m\}}\widetilde{\boldsymbol{A}}\boldsymbol{A}^{\{m-1\}}\right) - \sum_{k=r/2}^{r}\sum_{m=1}^{k} \theta_k \boldsymbol{W}^{\{k,m\}}$$

$$= \sum_{k=r/2}^{r} \theta_k \boldsymbol{Q}^{\{k\}} + \sum_{m=1}^{r} \left(\sum_{k=\max(r/2,m)}^{r} \theta_k \boldsymbol{Q}^{\{k-m\}}\widetilde{\boldsymbol{A}}\right) \boldsymbol{A}^{\{m-1\}} - \sum_{k=r/2}^{r}\sum_{m=1}^{k} \theta_k \boldsymbol{W}^{\{k,m\}}.$$

In particular, for any $\boldsymbol{u}$ that is orthogonal to $\boldsymbol{A}^{\{m\}}\mathbf{1}$ and $\boldsymbol{A}^{\{m\}}\boldsymbol{\nu}$ with norm-1 (i.e., a feasible vector of the variational definition in Lemma 7), we have:

$$\left\|\boldsymbol{\Gamma}_{\boldsymbol{\theta}}^{\{r\}}\boldsymbol{u}\right\|_2 \le \rho\left(\sum_{k=r/2}^{r} \theta_k \boldsymbol{Q}^{\{k\}}\right) + \sum_{m=1}^{r}\sum_{k=\max(r/2,m)}^{r} |\theta_k| \rho\left(\boldsymbol{Q}^{\{k-m\}}\right)\left\|\widetilde{\boldsymbol{A}}\boldsymbol{A}^{\{m-1\}}\boldsymbol{u}\right\|_2$$

$$+ \rho\left(\sum_{k=r/2}^{r} \theta_k \sum_{m=1}^{k} \boldsymbol{W}^{\{k,m\}}\right)$$

Since by (Massoulié, 2014), the terms $\rho\left(\boldsymbol{Q}^{\{k\}}\right)$ and $\rho\left(\boldsymbol{W}^{\{k,m\}}\right)$ are bounded by $n^\epsilon \xi_1^{r/2}$, $\left\|\widetilde{\boldsymbol{A}}\boldsymbol{A}^{\{m-1\}}\boldsymbol{x}\right\|_2$ is bounded by $O(\log(n) + \sqrt{\log(n)\xi_1^{m-1}})$, with some elementary calculations, we have that $\left\|\boldsymbol{\Gamma}_{\boldsymbol{\theta}}^{\{r\}}\boldsymbol{u}\right\|_2$ is bounded by $\|\boldsymbol{\theta}\|_1 n^\epsilon \xi_1^{r/2} O(\log(n))$.

Table A.1: Citation datasets. Label rate denotes the fraction of training labels in each dataset.

| Dataset | Nodes | Edges | Features | Classes | Label rates |
|---|---|---|---|---|---|
| Citeseer | 3,327 | 4,732 | 3,703 | 6 | 0.036 |
| Cora | 2,708 | 5,429 | 1,433 | 7 | 0.052 |
| Pubmed | 19,717 | 44,338 | 500 | 3 | 0.003 |

## A.5 MORE EXPERIMENTAL DETAILS AND RESULTS

**Experimental setup.** We followed the setup of (Yang et al., 2016; Kipf & Welling, 2017) for citation networks Citeseer, Cora and Pubmed. The statistics of datasets is in Table A.1. Degree

distribution in the original graph and the powered graphs for Citeseer and Cora is shown in Figure A.1. We used the same dataset splits as in (Yang et al., 2016; Kipf & Welling, 2017) with 20 labels per class for training, 500 labeled samples for hyper-parameter setting, and $1,000$ samples for testing. To facilitate comparison, we use the same hyper-parameters as Vanilla GCN (Kipf & Welling, 2017), including number of layers (2), number of hidden units (16), dropout rates (0.5), weight decay ($5 \times 10^{-4}$), and weight initialization following (Glorot & Bengio, 2010) implemented in Tensorflow. The training process is terminated if validation accuracy does not improve for 40 consecutive steps. We note that the citation datasets are extremely sensitive to initializations; as such, we report the test accuracy for the top 50 runs out of 100 random initializations sorted by the *validation* accuracy. We used Adam (Kingma & Ba, 2014) with a learning rate of 0.01 except for $\boldsymbol{\theta}$ (VPN), which was chosen to be $1 \times 10^{-5}$ to stabilize training. For VPN, we consider two-step training process, *i.e.*, we first employ graph sparsification on adjacency matrix $\bar{\boldsymbol{A}}^{[k]}$ based on $\boldsymbol{X}$ for the first-step training, then process second graph sparsification on adjacency matrix $\bar{\boldsymbol{A}}^{[k]}$ with respect to the feature embedding from hidden layer and fetch them back into model for second-step training with warm restart (continue training with the parameters from first-step). We set $\alpha_k$ to be 0.5 for the corresponding power order and 0 other wise in r-GCN.

To enhance robustness against evasion attack, we adaptively chose the threshold of graph sparsification. For each node, we consider all its neighbors within $r$ hops, where $r$ is the order of the powered graph. We order these neighbors based on their Euclidean distance to the current node in the feature space. We then set the threshold for this node such that only the first $s*d$ neighbors with the smallest distance are selected, where $s$ is the sparsification rate, and $d$ is the degree of the node in the original graph. If this node is a high-degree node (determined by if the degree is higher than the average degree by more than 2 standard deviations), we set $s = 1$; otherwise, we set $s$ to be a small number slightly larger than 1. Nevertheless, we do not observe that the robustness performance is sensitive to this parameter. In addition, one can define their own distance function that better measures proximity between nodes, such as correlation distance or cosine distance. As a side remark, one can easily obtain $\boldsymbol{A}^{(r)}$ with $\mathbb{1}[(\boldsymbol{I} + \boldsymbol{A})^r > 0]$, and recursively calculate $\boldsymbol{A}^{[k]}$. The proposed methods have similar computational complexity as vanilla GCN as a result of sparsification.

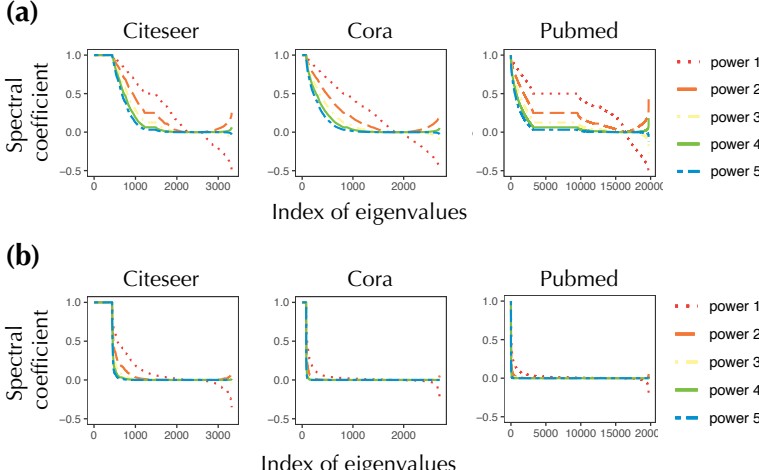

Figure A.2: Eigenspectrum of (a) the adjacency matrix and (b) VPN raised to different powers. The convolution operator becomes similar to low-pass filters as we increase powers. The eigenspectrum of VPN also has sharper edges than the original adjacency matrix.

**Revisiting SBM.** To obtain networks with different SNRs (0.58, 0.68, 0.79, 0.85, 0.91), we set the intra-degree average $a_{\text{intra}} = 2.1$ and inter-degree average $a_{\text{inter}}$ to be 0.4, 0.3, 0.2, 0.15, 0.1 respectively. We generate 10 networks for each SNR. One of the networks with SNR of 0.91 is analyzed in Figure A.3 for different convolution operators. As predicted by Theorem 3, the proposed operators have nontrivial spectral gap. This is *not* enjoyed by other operators, so it is a quite unique property. Furthermore, our learned operator w/ or w/o normalization obtained 81% accuracy based

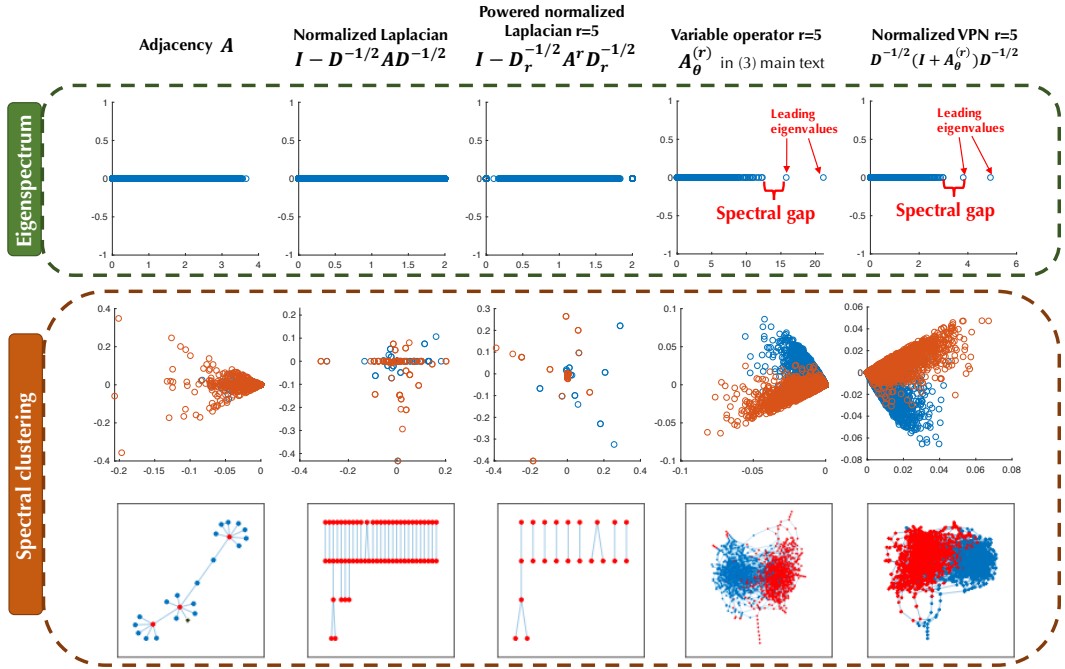

Figure A.3: Illustration of operators for SBM, showing (top to bottom) the eigenspectrum, plots of the first 2 leading eigenvectors, and the detected clusters. We consider (from left to right) the adjacency matrix $\boldsymbol{A}$, normalized Laplacian $\boldsymbol{I} - \boldsymbol{D}^{-1/2}\boldsymbol{A}\boldsymbol{D}^{-1/2}$, powered normalized Laplacian $\boldsymbol{I} - \boldsymbol{D}_r^{-1/2}\boldsymbol{A}^r\boldsymbol{D}_r^{-1/2}$ ($\boldsymbol{D}_r$ is the degree matrix of $\boldsymbol{A}^r$), VPN $\boldsymbol{A}_{\boldsymbol{\theta}}^{(r)}$ with order 5 ($\theta_k$ is 1 when $k = 1$ and 0.1 otherwise), and the normalized version $\boldsymbol{D}^{-1/2}(\boldsymbol{I} + \boldsymbol{A}_{\boldsymbol{\theta}}^{(r)})\boldsymbol{D}^{-1/2}$.

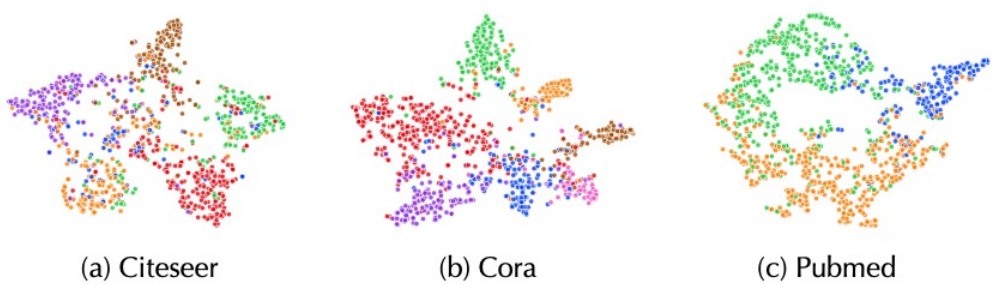

(a) Citeseer  (b) Cora  (c) Pubmed

Figure A.4: t-SNE visualization of network embeddings learned by r-GCN.

on the second eigenvector, whereas adjacency matrix catches "high-degree nodes" and normalized (or powered) Laplacians catch "tails", with spectral clustering accuracy of only 51%.

**Visualization of network embeddings.** We used t-SNE (Maaten & Hinton, 2008) to visualize the network embeddings learned by r-GCN (Figure A.4) and VPN (Figure A.5) for Citeseer, Cora and Pubmed. As can be seen, with very limited labels, the proposed methods can learn an effective embedding of the nodes.

**Evasion experiment.** Five representative global adversarial attack methods are considered as comparisons:

- DICE (Zügner & Günnemann, 2019) (remove internally, insert externally): it randomly disconnects nodes from the same class and connects nodes from different classes.

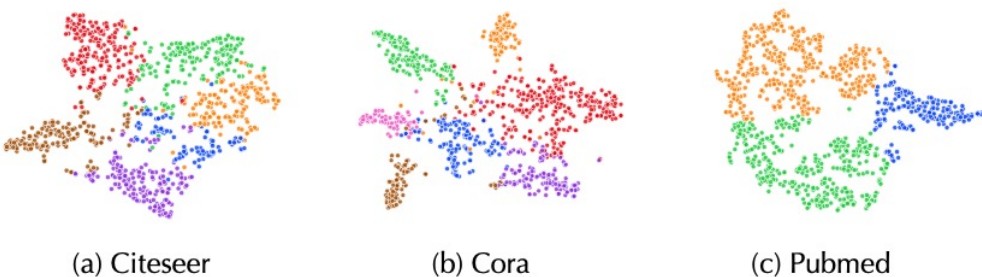

(a) Citeseer      (b) Cora      (c) Pubmed

Figure A.5: t-SNE visualization of network embeddings learned by VPN.

- $\mathcal{A}_{abr}$ and $\mathcal{A}_{DW_3}$ (Bojchevski & Günnemann, 2019): both attack methods are matrix perturbation theory based. $\mathcal{A}_{DW_3}$ is the closed-form attack for edge deletion and $\mathcal{A}_{abr}$ is the add-by-remove strategy based method for edge insertion.
- Meta-Train and Meta-Self (Zügner & Günnemann, 2019): both attack approaches are meta learning based. The meta-gradient approach with self-training is denoted as Meta-Self and the variant without self-training is denoted as Meta-Train. Note that both attack methods result in **OOM** (out of memory) on Pubmed, we choose to report the performance against them only on Citeseer and Cora.

We provide the post-evasion accuracy in Table A.2 and Table A.3 and robustness enhancement in Figure A.6, Figure A.7 and Figure A.8 for Citeseer, Cora, and Pubmed. Results are obtained with 20 random initializations on a given attacked network.

**Computational efficiency.** The adjacency matrix could be much denser after being powered several times, which poses a general challenge for all high-order matrix-based approaches. To alleviate this issue, we employ a simple sparsification strategy as described in the main paper. To demonstrate the computational efficiency of the proposed method, we introduce a running time comparison experiment on the real world social network (Social circles: Facebook) (Leskovec & Mcauley, 2012). This small dataset consists of "circles" (or "friends lists") from Facebook and becomes very dense from first order (4.71%) to second order (92.12%), thus is suitable to be used for computational efficiency evaluation. The results are shown in Table A.4. We can find that the running efficiency of the proposed method is compatible with baselines and the density does not affect the efficiency significantly when the dataset is not too large. Moreover, in this paper, our primary goal is to improve the robustness. We leave the solution to resolve the scalability issue as a future direction.

Table A.2: Summary of evasion attack performance on DICE, $\mathcal{A}_{abr}$ and $\mathcal{A}_{DW_3}$ in terms of post-evasion accuracy (in percent). Best performance under each setting is in bold.

| Attack Method | | | | | | DICE | | | | | |
|---|---|---|---|---|---|---|---|---|---|---|---|
| | | | | | | Model | | | | | |
| Attack Rate (%) | GCN | IGCN(RNM) | IGCN(AR) | LanczosNet | RGCN | PowerLap 2 | PowerLap 3 | SGC | MixHop | r-GCN | VPN |

**Citeseer**

| Attack Rate (%) | GCN | IGCN(RNM) | IGCN(AR) | LanczosNet | RGCN | PowerLap 2 | PowerLap 3 | SGC | MixHop | r-GCN | VPN |
|---|---|---|---|---|---|---|---|---|---|---|---|
| 5 | 68.5 ± 0.7 | 64.1 ± 0.0 | 64.4 ± 0.5 | 60.6 ± 0.9 | 69.9 ± 0.7 | 64.8 ± 0.4 | 65.9 ± 0.7 | 68.8 ± 0.5 | 69.9 ± 1.0 | **70.5 ± 1.0** | 70.0 ± 1.0 |
| 10 | 66.3 ± 0.1 | 61.6 ± 0.3 | 61.9 ± 0.2 | 57.6 ± 0.4 | 67.9 ± 0.5 | 62.5 ± 0.4 | 62.9 ± 0.4 | 67.5 ± 0.7 | 67.7 ± 0.7 | **68.6 ± 0.6** | 68.3 ± 0.9 |
| 15 | 63.4 ± 0.2 | 59.3 ± 0.5 | 59.4 ± 0.2 | 55.7 ± 0.5 | 66.0 ± 0.7 | 60.4 ± 0.4 | 60.2 ± 1.2 | 65.8 ± 0.7 | 65.4 ± 1.1 | 66.2 ± 0.4 | **66.4 ± 0.3** |
| 20 | 62.8 ± 0.4 | 56.9 ± 0.5 | 56.9 ± 0.2 | 53.0 ± 0.8 | 65.0 ± 1.0 | 57.8 ± 0.5 | 57.5 ± 0.7 | 64.7 ± 0.8 | 64.6 ± 1.0 | **65.4 ± 0.8** | 63.9 ± 0.9 |
| 25 | 60.9 ± 0.6 | 54.6 ± 0.3 | 54.7 ± 0.7 | 51.8 ± 0.4 | 63.2 ± 1.0 | 55.5 ± 0.4 | 55.3 ± 1.2 | 63.1 ± 0.5 | 63.2 ± 0.4 | **63.9 ± 0.0** | 63.2 ± 0.6 |
| 30 | 57.5 ± 1.1 | 51.9 ± 0.3 | 52.1 ± 0.7 | 49.7 ± 0.8 | 60.2 ± 0.5 | 52.4 ± 0.3 | 51.3 ± 1.9 | 59.2 ± 0.4 | 63.0 ± 0.7 | 59.6 ± 0.4 | **61.4 ± 0.4** |

**Cora**

| Attack Rate (%) | GCN | IGCN(RNM) | IGCN(AR) | LanczosNet | RGCN | PowerLap 2 | PowerLap 3 | SGC | MixHop | r-GCN | VPN |
|---|---|---|---|---|---|---|---|---|---|---|---|
| 5 | 79.2 ± 0.3 | 77.2 ± 0.9 | 78.0 ± 0.3 | 73.0 ± 0.9 | 79.5 ± 0.4 | 77.5 ± 1.1 | 74.0 ± 0.5 | 78.2 ± 0.2 | 80.1 ± 0.2 | **80.6 ± 0.8** | 78.4 ± 0.2 |
| 10 | 76.7 ± 0.7 | 73.7 ± 0.2 | 74.5 ± 0.7 | 69.6 ± 1.5 | 77.5 ± 0.5 | 74.6 ± 0.8 | 70.9 ± 0.5 | 76.1 ± 0.1 | 76.9 ± 0.1 | **78.6 ± 0.8** | 75.0 ± 0.3 |
| 15 | 74.3 ± 0.2 | 70.7 ± 0.1 | 71.5 ± 1.2 | 66.1 ± 0.0 | 74.6 ± 0.2 | 71.4 ± 1.1 | 67.0 ± 0.6 | 73.5 ± 1.7 | 73.1 ± 0.2 | **76.3 ± 0.8** | 72.6 ± 0.3 |
| 20 | 72.5 ± 0.6 | 69.5 ± 0.6 | 69.5 ± 0.6 | 62.8 ± 0.7 | 72.9 ± 0.2 | 69.9 ± 0.7 | 64.0 ± 0.3 | 72.1 ± 0.3 | 71.6 ± 1.0 | **74.6 ± 0.8** | 71.3 ± 0.3 |
| 25 | 70.2 ± 0.9 | 66.2 ± 0.5 | 67.4 ± 0.9 | 60.4 ± 1.3 | 70.9 ± 0.4 | 66.6 ± 1.1 | 59.8 ± 0.7 | 70.1 ± 0.3 | 69.5 ± 0.6 | **72.7 ± 0.9** | 69.4 ± 0.3 |
| 30 | 69.9 ± 0.4 | 67.5 ± 0.8 | 68.9 ± 1.4 | 58.7 ± 1.1 | 72.2 ± 0.6 | 66.7 ± 0.8 | 58.8 ± 0.8 | 70.8 ± 0.6 | 69.7 ± 0.3 | **72.9 ± 0.8** | 69.7 ± 0.5 |

**Pubmed**

| Attack Rate (%) | GCN | IGCN(RNM) | IGCN(AR) | LanczosNet | RGCN | PowerLap 2 | PowerLap 3 | SGC | MixHop | r-GCN | VPN |
|---|---|---|---|---|---|---|---|---|---|---|---|
| 5 | 75.3 ± 0.4 | 74.8 ± 0.8 | 75.8 ± 0.2 | 72.9 ± 1.3 | 76.3 ± 0.4 | 74.9 ± 0.9 | 68.0 ± 2.7 | 75.6 ± 0.5 | 76.6 ± 0.5 | 76.1 ± 0.5 | **77.1 ± 0.6** |
| 10 | 73.0 ± 0.5 | 72.3 ± 0.5 | 73.0 ± 0.2 | 69.9 ± 1.0 | 73.9 ± 0.4 | 72.1 ± 0.9 | 65.4 ± 2.7 | 73.1 ± 0.2 | 73.0 ± 0.8 | 73.6 ± 0.3 | **74.4 ± 0.1** |
| 15 | 71.6 ± 0.1 | 68.8 ± 0.2 | 70.4 ± 0.2 | 67.4 ± 0.1 | 70.9 ± 0.2 | 69.3 ± 1.0 | 62.5 ± 2.0 | 79.4 ± 0.2 | 70.7 ± 0.7 | 70.7 ± 0.1 | **71.3 ± 0.9** |
| 20 | 69.2 ± 0.6 | 66.1 ± 0.2 | 67.6 ± 0.4 | 63.7 ± 1.8 | 69.2 ± 0.2 | 67.5 ± 0.8 | 60.5 ± 1.3 | 69.1 ± 0.7 | 67.0 ± 0.8 | 69.8 ± 0.3 | **70.1 ± 0.7** |
| 25 | 68.2 ± 0.6 | 63.6 ± 0.3 | 65.7 ± 0.2 | 61.5 ± 1.6 | 67.8 ± 0.5 | 64.4 ± 0.5 | 58.0 ± 1.2 | 67.3 ± 0.2 | 64.5 ± 0.2 | 68.3 ± 0.3 | **68.8 ± 0.1** |
| 30 | 57.5 ± 0.9 | 63.7 ± 0.2 | 65.0 ± 1.0 | 62.2 ± 0.3 | 66.3 ± 0.2 | 63.5 ± 0.7 | 56.3 ± 1.0 | 66.0 ± 0.5 | 64.5 ± 0.3 | 66.5 ± 0.1 | **67.1 ± 0.3** |

| Attack Method | | | | | | $\mathcal{A}_{abr}$ | | | | | |
|---|---|---|---|---|---|---|---|---|---|---|---|
| | | | | | | Model | | | | | |
| Attack Rate (%) | GCN | IGCN(RNM) | IGCN(AR) | LanczosNet | RGCN | PowerLap 2 | PowerLap 3 | SGC | MixHop | r-GCN | VPN |

**Citeseer**

| Attack Rate (%) | GCN | IGCN(RNM) | IGCN(AR) | LanczosNet | RGCN | PowerLap 2 | PowerLap 3 | SGC | MixHop | r-GCN | VPN |
|---|---|---|---|---|---|---|---|---|---|---|---|
| 5 | 69.2 ± 0.2 | 61.7 ± 0.1 | 60.4 ± 0.1 | 57.3 ± 0.7 | 70.4 ± 0.8 | 68.3 ± 0.2 | 64.1 ± 2.0 | 70.7 ± 0.2 | 70.4 ± 0.8 | 70.8 ± 0.5 | **71.6 ± 0.6** |
| 10 | 67.8 ± 0.3 | 59.1 ± 0.3 | 56.3 ± 0.3 | 55.0 ± 0.2 | 70.2 ± 0.8 | 66.0 ± 0.2 | 59.0 ± 2.7 | 69.6 ± 0.3 | 69.4 ± 0.9 | 70.6 ± 0.0 | **70.8 ± 1.0** |
| 15 | 66.6 ± 0.4 | 53.6 ± 0.4 | 52.1 ± 0.4 | 50.7 ± 0.9 | 70.2 ± 1.0 | 63.9 ± 0.2 | 55.3 ± 2.3 | 69.1 ± 0.3 | 68.7 ± 0.4 | 70.4 ± 0.4 | **70.6 ± 0.7** |
| 20 | 66.4 ± 0.5 | 49.7 ± 0.5 | 49.5 ± 0.5 | 47.4 ± 0.8 | 69.9 ± 0.8 | 62.0 ± 0.3 | 52.2 ± 2.2 | 69.5 ± 0.5 | 67.9 ± 1.2 | **70.3 ± 0.2** | 70.1 ± 0.3 |
| 25 | 65.6 ± 0.5 | 45.4 ± 0.6 | 45.3 ± 0.6 | 44.0 ± 1.0 | 69.1 ± 1.0 | 59.5 ± 0.3 | 48.6 ± 1.5 | 68.6 ± 0.4 | 67.7 ± 2.1 | **70.0 ± 0.9** | 69.7 ± 0.4 |
| 30 | 63.8 ± 0.5 | 41.9 ± 0.6 | 42.6 ± 0.6 | 40.5 ± 0.9 | 68.7 ± 1.0 | 57.2 ± 0.4 | 44.8 ± 1.4 | 67.4 ± 0.3 | 67.3 ± 0.5 | **69.5 ± 0.4** | 69.1 ± 0.6 |

**Cora**

| Attack Rate (%) | GCN | IGCN(RNM) | IGCN(AR) | LanczosNet | RGCN | PowerLap 2 | PowerLap 3 | SGC | MixHop | r-GCN | VPN |
|---|---|---|---|---|---|---|---|---|---|---|---|
| 5 | 81.1 ± 0.1 | 77.6 ± 0.1 | 80.5 ± 0.1 | 75.3 ± 1.0 | 81.0 ± 0.4 | 79.8 ± 0.4 | 75.2 ± 1.1 | 79.7 ± 0.6 | 81.1 ± 0.3 | 80.9 ± 0.9 | **81.5 ± 0.2** |
| 10 | 80.3 ± 0.1 | 76.4 ± 0.2 | 79.3 ± 0.1 | 74.0 ± 1.7 | 80.0 ± 0.6 | 78.9 ± 0.7 | 73.0 ± 1.5 | 78.7 ± 0.8 | 80.6 ± 0.3 | 79.6 ± 0.8 | **81.1 ± 0.8** |
| 15 | 79.7 ± 0.2 | 74.3 ± 0.4 | 77.7 ± 0.2 | 72.7 ± 1.0 | 80.1 ± 0.4 | 77.1 ± 1.2 | 69.9 ± 1.7 | 78.6 ± 0.1 | 79.7 ± 1.5 | 79.7 ± 0.1 | **80.0 ± 0.5** |
| 20 | 78.9 ± 0.2 | 71.2 ± 0.5 | 73.2 ± 0.2 | 70.5 ± 0.2 | 79.2 ± 0.7 | 75.9 ± 1.6 | 66.8 ± 2.5 | 78.2 ± 0.4 | 79.0 ± 0.4 | 79.1 ± 0.2 | **79.3 ± 0.6** |
| 25 | 77.4 ± 0.3 | 64.9 ± 0.6 | 70.1 ± 0.3 | 68.2 ± 0.6 | 78.5 ± 1.1 | 73.2 ± 1.4 | 61.7 ± 2.6 | 77.1 ± 0.3 | 78.0 ± 0.3 | **78.5 ± 0.1** | 78.4 ± 0.6 |
| 30 | 75.6 ± 0.6 | 59.2 ± 0.7 | 66.2 ± 0.4 | 65.5 ± 0.8 | 77.7 ± 0.8 | 71.0 ± 1.9 | 57.4 ± 2.8 | 76.0 ± 0.3 | 77.1 ± 0.6 | 77.3 ± 0.8 | **77.6 ± 1.0** |

**Pubmed**

| Attack Rate (%) | GCN | IGCN(RNM) | IGCN(AR) | LanczosNet | RGCN | PowerLap 2 | PowerLap 3 | SGC | MixHop | r-GCN | VPN |
|---|---|---|---|---|---|---|---|---|---|---|---|
| 5 | 75.3 ± 0.8 | 76.4 ± 0.2 | 75.9 ± 0.4 | 75.0 ± 0.2 | 77.6 ± 0.1 | 76.9 ± 0.5 | 67.3 ± 1.2 | 77.8 ± 0.8 | 78.2 ± 0.3 | 78.2 ± 0.3 | **78.6 ± 0.6** |
| 10 | 74.3 ± 0.5 | 74.3 ± 0.1 | 73.0 ± 0.6 | 72.7 ± 0.8 | 77.0 ± 0.1 | 75.6 ± 0.6 | 64.9 ± 1.7 | 77.5 ± 0.9 | 77.2 ± 2.2 | 77.7 ± 0.5 | **77.7 ± 0.4** |
| 15 | 73.5 ± 0.7 | 71.9 ± 0.4 | 69.4 ± 0.0 | 70.9 ± 0.6 | 76.1 ± 0.3 | 74.1 ± 0.3 | 61.9 ± 1.8 | 76.5 ± 0.4 | 76.5 ± 0.5 | 76.9 ± 0.4 | **77.3 ± 0.8** |
| 20 | 73.7 ± 0.1 | 68.7 ± 0.7 | 66.2 ± 0.0 | 69.1 ± 0.8 | 76.5 ± 0.1 | 73.7 ± 0.3 | 59.4 ± 1.1 | 76.3 ± 1.1 | 77.3 ± 1.2 | 77.3 ± 0.4 | **77.7 ± 0.1** |
| 25 | 72.9 ± 0.1 | 64.9 ± 0.2 | 63.2 ± 0.5 | 67.6 ± 0.9 | 75.8 ± 0.3 | 73.1 ± 0.5 | 58.0 ± 0.6 | 75.5 ± 0.4 | 75.7 ± 0.4 | 76.7 ± 0.7 | **77.2 ± 0.3** |
| 30 | 72.8 ± 0.9 | 63.1 ± 0.6 | 61.1 ± 0.6 | 65.8 ± 1.2 | 75.9 ± 0.2 | 72.4 ± 0.6 | 55.4 ± 0.6 | 76.2 ± 0.3 | 75.9 ± 0.5 | **77.2 ± 0.6** | 76.7 ± 0.5 |

| Attack Method | | | | | | $\mathcal{A}_{DW_3}$ | | | | | |
|---|---|---|---|---|---|---|---|---|---|---|---|
| | | | | | | Model | | | | | |
| Attack Rate (%) | GCN | IGCN(RNM) | IGCN(AR) | LanczosNet | RGCN | PowerLap 2 | PowerLap 3 | SGC | MixHop | r-GCN | VPN |

**Citeseer**

| Attack Rate (%) | GCN | IGCN(RNM) | IGCN(AR) | LanczosNet | RGCN | PowerLap 2 | PowerLap 3 | SGC | MixHop | r-GCN | VPN |
|---|---|---|---|---|---|---|---|---|---|---|---|
| 5 | 68.8 ± 0.1 | 63.5 ± 0.4 | 65.1 ± 0.2 | 60.0 ± 0.5 | 70.3 ± 0.6 | 69.2 ± 1.0 | 70.0 ± 0.8 | 70.5 ± 0.5 | 70.4 ± 1.0 | **71.8 ± 0.3** | 70.6 ± 0.6 |
| 10 | 68.4 ± 0.2 | 63.1 ± 0.6 | 63.6 ± 0.5 | 59.6 ± 0.6 | 70.1 ± 0.6 | 69.3 ± 1.1 | 69.6 ± 0.6 | 70.1 ± 0.8 | 69.4 ± 0.4 | **71.2 ± 0.7** | 70.2 ± 0.7 |
| 15 | 68.9 ± 0.1 | 62.9 ± 0.7 | 63.2 ± 0.8 | 59.5 ± 0.9 | 70.3 ± 0.4 | 69.5 ± 1.1 | 69.8 ± 0.7 | 70.6 ± 0.8 | 68.7 ± 1.4 | **71.2 ± 0.1** | 70.6 ± 0.4 |
| 20 | 68.8 ± 0.5 | 63.5 ± 0.8 | 63.4 ± 0.7 | 59.1 ± 0.8 | 69.8 ± 0.4 | 69.4 ± 1.0 | 69.5 ± 0.7 | 70.3 ± 0.2 | 67.9 ± 1.4 | **71.1 ± 0.2** | 70.4 ± 0.7 |
| 25 | 68.8 ± 0.4 | 63.6 ± 0.7 | 63.8 ± 0.6 | 59.3 ± 0.4 | 69.9 ± 0.5 | 69.2 ± 0.9 | 69.3 ± 0.9 | 70.2 ± 0.5 | 67.7 ± 0.8 | **71.2 ± 0.1** | 70.1 ± 0.2 |
| 30 | 68.8 ± 0.4 | 63.6 ± 0.7 | 63.8 ± 0.6 | 59.3 ± 0.1 | 69.9 ± 0.5 | 69.2 ± 0.9 | 69.3 ± 0.9 | 70.2 ± 0.7 | 67.3 ± 0.1 | **71.2 ± 0.0** | 70.1 ± 0.9 |

**Cora**

| Attack Rate (%) | GCN | IGCN(RNM) | IGCN(AR) | LanczosNet | RGCN | PowerLap 2 | PowerLap 3 | SGC | MixHop | r-GCN | VPN |
|---|---|---|---|---|---|---|---|---|---|---|---|
| 5 | 80.0 ± 0.7 | 78.9 ± 0.2 | 79.2 ± 0.4 | 78.1 ± 0.8 | 80.5 ± 0.9 | 77.8 ± 0.9 | 77.0 ± 1.4 | 78.3 ± 0.3 | 80.3 ± 0.3 | 80.8 ± 0.7 | **80.8 ± 0.5** |
| 10 | 79.7 ± 0.9 | 78.7 ± 0.2 | 78.9 ± 0.4 | 77.8 ± 1.0 | 80.1 ± 0.6 | 78.1 ± 1.1 | 77.8 ± 1.1 | 78.0 ± 0.9 | 80.1 ± 0.2 | **80.7 ± 0.8** | 80.5 ± 0.3 |
| 15 | 79.3 ± 0.7 | 77.1 ± 0.2 | 77.7 ± 0.4 | 76.8 ± 1.2 | 79.7 ± 0.5 | 77.3 ± 1.2 | 77.9 ± 0.6 | 77.8 ± 0.5 | 80.0 ± 0.7 | 80.3 ± 0.9 | **80.6 ± 0.9** |
| 20 | 78.3 ± 1.0 | 75.5 ± 0.2 | 75.6 ± 0.3 | 75.7 ± 0.3 | 79.3 ± 0.6 | 76.9 ± 1.2 | 77.1 ± 0.5 | 78.0 ± 0.2 | 79.8 ± 1.0 | 80.3 ± 0.5 | **80.3 ± 0.2** |
| 25 | 77.8 ± 0.6 | 75.3 ± 0.3 | 74.4 ± 0.7 | 74.6 ± 1.8 | 78.7 ± 0.5 | 76.1 ± 1.0 | 76.8 ± 0.2 | 77.3 ± 0.2 | 79.1 ± 0.7 | **79.2 ± 0.0** | 79.1 ± 0.5 |
| 30 | 77.8 ± 0.8 | 75.0 ± 0.2 | 73.8 ± 0.7 | 74.3 ± 1.7 | 78.0 ± 0.5 | 75.6 ± 1.1 | 75.9 ± 0.6 | 76.7 ± 0.6 | 78.8 ± 0.4 | 79.3 ± 0.9 | **78.6 ± 1.0** |

**Pubmed**

| Attack Rate (%) | GCN | IGCN(RNM) | IGCN(AR) | LanczosNet | RGCN | PowerLap 2 | PowerLap 3 | SGC | MixHop | r-GCN | VPN |
|---|---|---|---|---|---|---|---|---|---|---|---|
| 5 | 78.3 ± 0.5 | 79.3 ± 0.4 | 79.2 ± 0.1 | 76.5 ± 1.8 | 77.8 ± 0.1 | 74.8 ± 1.7 | 76.5 ± 0.5 | 78.0 ± 0.4 | 78.2 ± 0.5 | 78.1 ± 0.7 | **79.0 ± 1.0** |
| 10 | 77.8 ± 0.2 | 78.6 ± 0.2 | 78.6 ± 0.7 | 75.3 ± 1.0 | 77.9 ± 0.3 | 74.7 ± 1.9 | 77.4 ± 0.4 | 77.8 ± 1.0 | 78.0 ± 0.7 | 77.9 ± 0.9 | **78.6 ± 0.1** |
| 15 | 77.4 ± 0.4 | 77.0 ± 0.4 | 77.2 ± 0.3 | 74.7 ± 0.3 | 77.5 ± 0.3 | 74.2 ± 1.5 | 76.9 ± 0.4 | 77.3 ± 0.4 | 77.3 ± 0.8 | 77.8 ± 0.3 | **78.3 ± 0.2** |
| 20 | 77.4 ± 0.4 | 75.8 ± 0.3 | 76.3 ± 0.4 | 73.5 ± 1.8 | 77.8 ± 0.4 | 73.8 ± 1.0 | 76.3 ± 0.4 | 77.3 ± 1.2 | 77.1 ± 0.3 | 77.6 ± 1.0 | **78.3 ± 0.4** |
| 25 | 77.2 ± 0.5 | 75.2 ± 0.1 | 75.9 ± 0.4 | 73.7 ± 1.1 | 77.6 ± 0.3 | 73.7 ± 1.1 | 75.6 ± 0.7 | 77.1 ± 0.2 | 77.0 ± 0.3 | 78.1 ± 0.8 | **78.2 ± 0.2** |
| 30 | 77.1 ± 0.2 | 74.5 ± 0.3 | 74.6 ± 0.3 | 73.4 ± 0.3 | 77.0 ± 0.2 | 72.9 ± 1.4 | 74.6 ± 0.4 | 77.1 ± 0.3 | 77.2 ± 0.8 | 77.6 ± 0.3 | **77.7 ± 0.9** |

Table A.3: Summary of evasion attack performance on Meta-Train and Meta-Self in terms of post-evasion accuracy (in percent). **OOM** (out of memory) for both methods on Pubmed.

| Attack Method | | | | | | Meta-Train | | | | | | |
| | | | | | | Model | | | | | | |
| | Attack Rate (%) | GCN | IGCN(RNM) | IGCN(AR) | LanczosNet | RGCN | PowerLap 2 | PowerLap 3 | SGC | MixHop | r-GCN | VPN |
|---|---|---|---|---|---|---|---|---|---|---|---|---|
| Citeseer | 5 | 67.6 ± 0.3 | 62.8 ± 0.8 | 63.1 ± 1.2 | 57.7 ± 0.7 | 69.3 ± 0.5 | 66.8 ± 1.1 | 64.9 ± 1.5 | 70.0 ± 0.3 | 69.6 ± 0.6 | 70.4 ± 0.7 | **70.4 ± 0.6** |
| | 10 | 66.0 ± 1.0 | 61.0 ± 1.1 | 61.4 ± 1.0 | 56.4 ± 0.3 | 68.3 ± 0.7 | 65.2 ± 0.9 | 61.9 ± 1.4 | 68.3 ± 0.2 | 68.9 ± 1.0 | 68.7 ± 0.4 | **69.4 ± 1.0** |
| | 15 | 66.1 ± 0.6 | 60.9 ± 1.1 | 61.4 ± 1.0 | 55.4 ± 0.8 | 67.3 ± 0.4 | 64.5 ± 1.3 | 60.9 ± 1.9 | 67.3 ± 0.8 | 67.8 ± 0.8 | 67.9 ± 0.6 | **68.6 ± 0.3** |
| | 20 | 64.9 ± 0.9 | 59.1 ± 1.4 | 59.7 ± 0.8 | 54.5 ± 0.1 | 66.6 ± 0.6 | 63.0 ± 1.2 | 58.2 ± 2.1 | 67.2 ± 0.5 | 67.1 ± 0.9 | 67.6 ± 0.4 | **68.5 ± 0.8** |
| | 25 | 64.3 ± 0.7 | 57.9 ± 1.8 | 59.8 ± 1.0 | 54.0 ± 0.4 | 66.3 ± 0.7 | 62.7 ± 1.1 | 55.8 ± 2.6 | 66.1 ± 0.5 | 66.5 ± 1.7 | 67.0 ± 0.5 | **67.0 ± 0.2** |
| | 30 | 64.8 ± 1.1 | 59.8 ± 1.0 | 59.0 ± 0.8 | 53.5 ± 0.8 | 65.2 ± 0.4 | 61.8 ± 0.8 | 56.1 ± 1.9 | 65.0 ± 0.8 | 65.5 ± 0.7 | 66.0 ± 0.8 | **66.1 ± 0.1** |
| | Attack Rate (%) | GCN | IGCN(RNM) | IGCN(AR) | LanczosNet | RGCN | PowerLap 2 | PowerLap 3 | SGC | MixHop | r-GCN | VPN |
| Cora | 5 | 79.8 ± 0.9 | 76.5 ± 0.1 | 77.4 ± 0.3 | 75.1 ± 1.1 | 80.2 ± 0.7 | 75.2 ± 1.2 | 75.1 ± 1.1 | 79.5 ± 0.5 | 80.5 ± 0.8 | 79.8 ± 0.6 | **80.8 ± 0.2** |
| | 10 | 78.4 ± 0.7 | 75.8 ± 0.4 | 76.6 ± 0.3 | 73.3 ± 1.7 | 79.0 ± 0.5 | 74.2 ± 1.2 | 72.9 ± 1.0 | 78.1 ± 0.7 | 79.0 ± 0.1 | 78.5 ± 0.9 | **79.2 ± 0.1** |
| | 15 | 77.9 ± 0.6 | 73.8 ± 0.9 | 76.7 ± 0.6 | 71.8 ± 1.1 | 79.0 ± 0.7 | 73.4 ± 1.7 | 71.0 ± 1.1 | 77.9 ± 1.8 | 78.3 ± 0.6 | 78.2 ± 0.6 | **79.7 ± 0.1** |
| | 20 | 76.0 ± 1.4 | 71.8 ± 1.0 | 73.8 ± 0.3 | 70.2 ± 1.9 | 77.1 ± 0.5 | 71.0 ± 1.5 | 68.4 ± 1.0 | 76.3 ± 1.0 | 77.2 ± 0.2 | 76.6 ± 0.9 | **77.6 ± 1.0** |
| | 25 | 79.1 ± 0.6 | 77.7 ± 0.6 | 78.7 ± 0.6 | 74.3 ± 0.7 | 79.7 ± 0.6 | 74.3 ± 1.6 | 73.8 ± 1.0 | 78.0 ± 0.4 | 77.7 ± 1.0 | 78.9 ± 1.0 | **79.8 ± 0.9** |
| | 30 | 77.7 ± 0.9 | 75.6 ± 0.7 | 76.7 ± 0.6 | 73.8 ± 0.7 | 78.6 ± 0.4 | 73.5 ± 1.6 | 73.4 ± 0.7 | 78.3 ± 0.4 | 78.4 ± 0.9 | 78.6 ± 0.9 | **79.0 ± 0.0** |

| Attack Method | | | | | | Meta-Self | | | | | | |
| | | | | | | Model | | | | | | |
| | Attack Rate (%) | GCN | IGCN(RNM) | IGCN(AR) | LanczosNet | RGCN | PowerLap 2 | PowerLap 3 | SGC | MixHop | r-GCN | VPN |
|---|---|---|---|---|---|---|---|---|---|---|---|---|
| Citeseer | 5 | 66.2 ± 0.9 | 61.8 ± 0.1 | 60.9 ± 0.8 | 60.2 ± 0.2 | 69.1 ± 0.4 | 66.3 ± 0.9 | 66.0 ± 1.2 | 69.1 ± 0.3 | 69.6 ± 0.4 | **70.1 ± 0.7** | 69.1 ± 0.8 |
| | 10 | 64.7 ± 0.8 | 59.3 ± 0.4 | 59.1 ± 0.9 | 57.3 ± 0.1 | 67.3 ± 0.6 | 64.4 ± 0.8 | 62.5 ± 1.3 | 67.9 ± 0.7 | 67.1 ± 0.5 | 68.0 ± 0.0 | **68.2 ± 0.7** |
| | 15 | 64.0 ± 0.9 | 57.2 ± 0.3 | 58.1 ± 0.7 | 55.2 ± 0.2 | 65.9 ± 0.9 | 62.9 ± 1.7 | 60.6 ± 0.7 | 66.3 ± 0.9 | 66.2 ± 0.4 | **67.3 ± 0.9** | 66.2 ± 0.1 |
| | 20 | 62.4 ± 1.2 | 55.5 ± 0.2 | 56.8 ± 0.8 | 55.9 ± 0.4 | 65.3 ± 1.2 | 61.2 ± 1.5 | 58.6 ± 1.9 | 65.3 ± 1.0 | 64.9 ± 0.1 | **66.3 ± 0.4** | 65.6 ± 0.4 |
| | 25 | 62.7 ± 1.0 | 56.6 ± 0.7 | 58.4 ± 1.1 | 55.3 ± 0.3 | 65.1 ± 0.7 | 60.5 ± 1.6 | 57.6 ± 1.9 | 65.6 ± 0.8 | 65.0 ± 0.6 | **65.9 ± 0.5** | 65.5 ± 0.5 |
| | 30 | 61.1 ± 0.9 | 55.9 ± 0.5 | 56.4 ± 0.7 | 54.9 ± 0.1 | 64.7 ± 0.3 | 60.3 ± 1.5 | 57.5 ± 1.4 | 64.1 ± 0.8 | 64.5 ± 1.2 | **66.0 ± 1.0** | 64.5 ± 0.9 |
| | Attack Rate (%) | GCN | IGCN(RNM) | IGCN(AR) | LanczosNet | RGCN | PowerLap 2 | PowerLap 3 | SGC | MixHop | r-GCN | VPN |
| Cora | 5 | 78.7 ± 0.2 | 76.1 ± 0.3 | 75.0 ± 0.4 | 73.4 ± 0.5 | 79.6 ± 0.6 | 77.9 ± 1.4 | 74.7 ± 0.7 | 77.9 ± 0.6 | 79.0 ± 0.8 | **79.3 ± 0.8** | 79.1 ± 0.5 |
| | 10 | 76.8 ± 0.3 | 76.0 ± 0.5 | 74.3 ± 0.7 | 71.0 ± 1.8 | 78.4 ± 0.3 | 76.0 ± 0.9 | 72.6 ± 0.6 | 76.8 ± 0.2 | 77.2 ± 0.1 | 77.8 ± 0.9 | **78.0 ± 0.7** |
| | 15 | 75.3 ± 0.9 | 74.0 ± 0.5 | 72.1 ± 0.4 | 69.4 ± 0.0 | 77.1 ± 0.7 | 74.2 ± 1.4 | 69.5 ± 1.2 | 76.6 ± 0.4 | 77.0 ± 0.6 | 77.0 ± 0.7 | **77.5 ± 0.9** |
| | 20 | 75.8 ± 0.7 | 72.6 ± 0.7 | 71.1 ± 0.8 | 69.5 ± 1.4 | 76.3 ± 0.5 | 73.3 ± 1.2 | 70.4 ± 1.4 | 75.9 ± 0.3 | 76.3 ± 0.4 | 76.1 ± 0.9 | **76.4 ± 0.5** |
| | 25 | 77.5 ± 0.2 | 73.8 ± 0.2 | 73.4 ± 0.6 | 70.3 ± 0.2 | 78.4 ± 0.4 | 74.2 ± 1.0 | 70.9 ± 1.0 | 76.9 ± 0.3 | 77.6 ± 0.2 | **77.9 ± 0.2** | 77.8 ± 0.1 |
| | 30 | 76.7 ± 0.6 | 73.7 ± 0.6 | 72.2 ± 1.0 | 66.7 ± 0.0 | 78.4 ± 0.5 | 72.6 ± 1.2 | 67.0 ± 1.4 | 77.9 ± 0.2 | 78.0 ± 0.7 | 78.1 ± 0.5 | **78.5 ± 0.4** |

Table A.4: Running time comparison on a social network dataset (Leskovec & Mcauley, 2012).

| Methods | Vanilla GCN | PowerLaplacian | IGCN(RNM) | IGCN(AR) | LNet | RGCN | SGC | MixHop | **r-GCN** | **VPN** |
|---|---|---|---|---|---|---|---|---|---|---|
| Running Time (sec) | 6.02 | 6.36 | 3.33 | 7.21 | 5.56 | 18.14 | 0.231 | 11.38 | 6.73 | 7.18 |

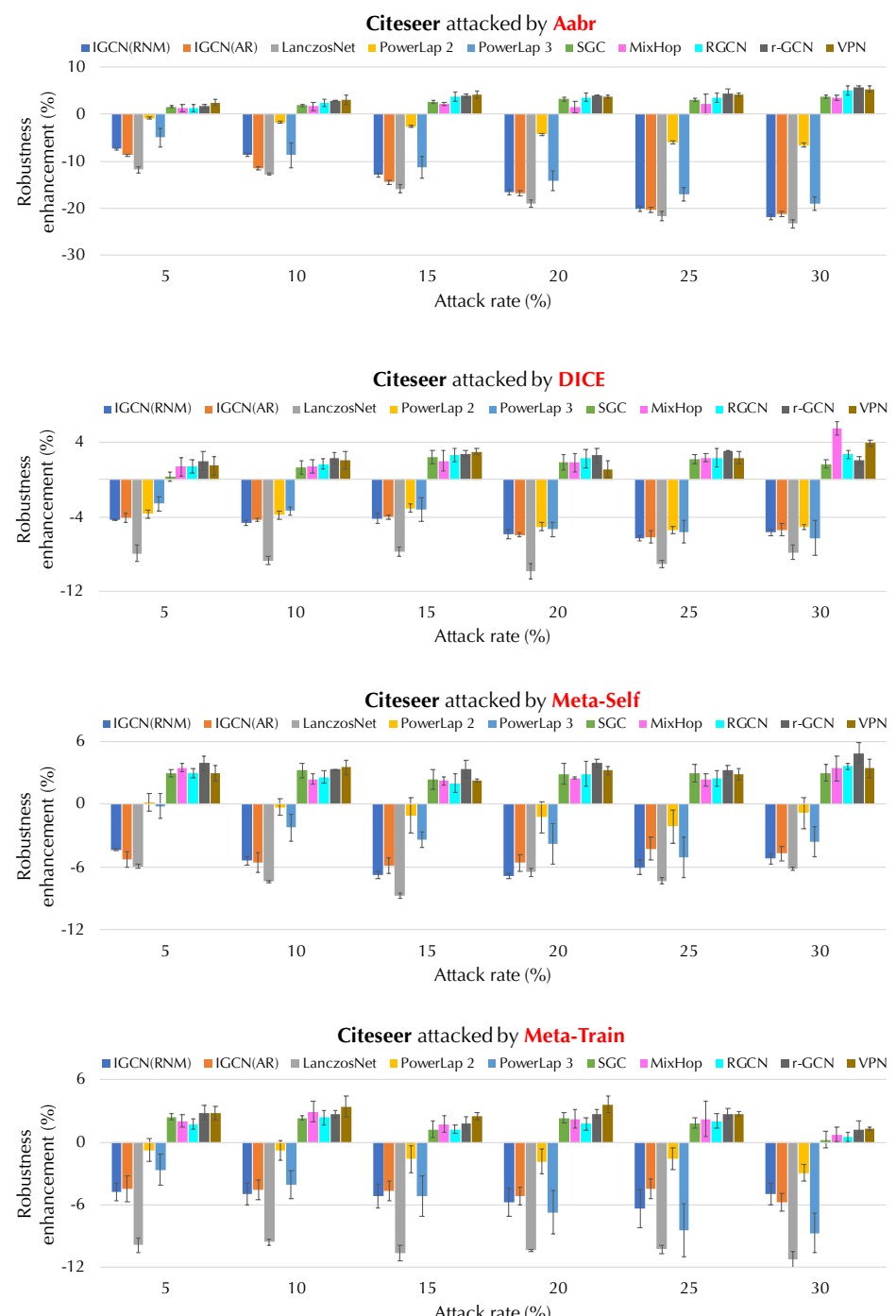

Figure A.6: Illustration of evasion attack performance in terms of robustness enhancement on different attack methods (from top to bottom) for **Citeseer**: $\mathcal{A}_{abr}$ (Bojchevski & Günnemann, 2019), DICE (Zügner & Günnemann, 2019), Meta-Self and Meta-Train (Zügner & Günnemann, 2019). Evaluated methods include IGCN (Li et al., 2019), LanczosNet (Liao et al., 2019), SGC (Wu et al., 2019), MixHop (Abu-El-Haija et al., 2019) and RGCN (Zhu et al., 2019). PowerLap 2 and 3 are methods that replace the adjacency matrix in GCN by its powered versions with orders 2 and 3, respectively. Positive values indicate improvement of robustness compared to vanilla GCN and negative ones indicate deterioration.

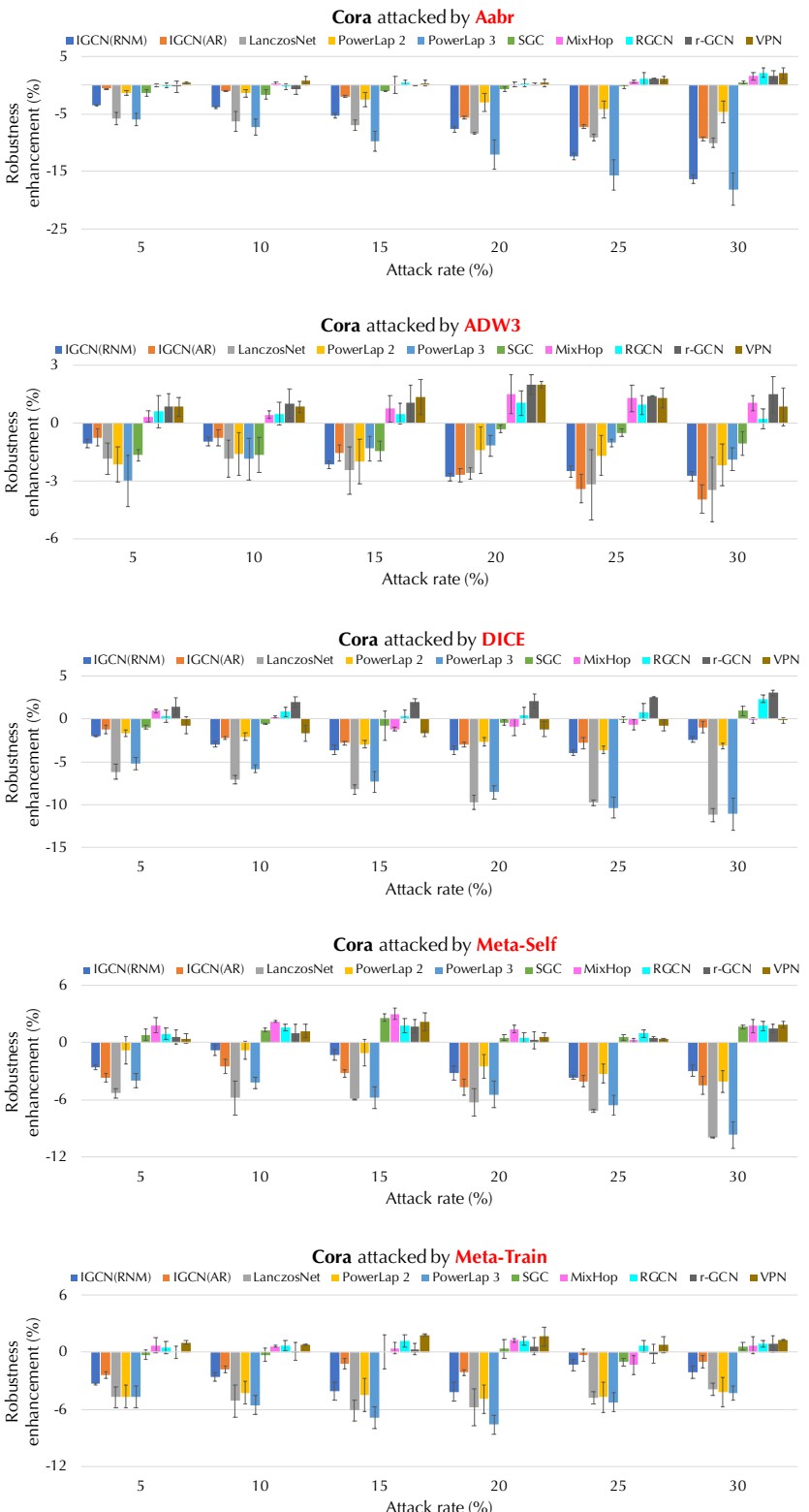

Figure A.7: Illustration of evasion attack performance in terms of robustness enhancement on different attack methods for **Cora**.

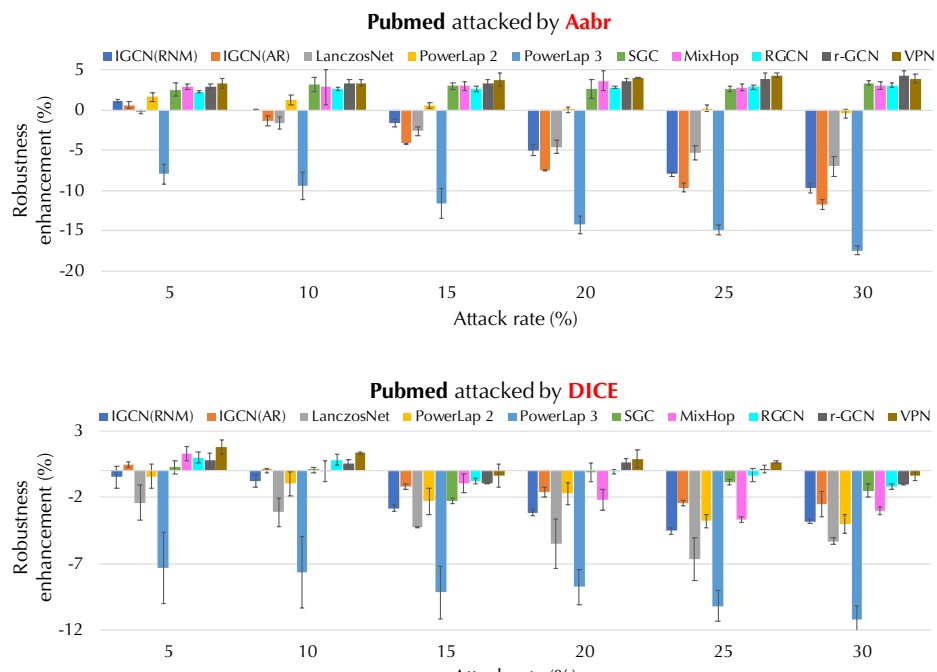

Figure A.8: Illustration of evasion attack performance in terms of robustness enhancement on different attack methods for **Pubmed**.

