# OpenReview forum: "Power up! Robust Graph Convolutional Network based on Graph Powering"
_ICLR.cc/2020/Conference — Reject_

### Official Review · AnonReviewer2 · 2019-10-08
**Official Blind Review #2**

**Rating:** 3

**Review:**

This paper proposes a new architecture for graph convolutional network based on graph powering operation which generates a new graph based on the shortest distance between pair of nodes.  Its main motivation is to overcome the dominance of the first eigenvector in the existing GCN architectures based on the graph Laplacian operator. The theoretical evidence for the robustness is provided based on the signal-to-noise (SNR) ratio of the simplified stochastic block model (SBM). Two versions of the algorithms are proposed, namely the robust graph convolutional network (r-GCN) and variable power network (VPN). First, r-GCN is based on augmenting the graphs with graph powering operation. Next, VPN replaces the adjacency matrix of the graph convolutional operator by the newly proposed variable power operator. An additional sparsification scheme is proposed since the graph powering operation densifies the original graph.

Overall, I like how the paper addresses the weakness of the existing graph Laplacian operators (dominance of the first eigenvector) and proposed a new method with theoretical justifications. Experiments were conducted thoroughly and results look great in the presented datasets. However, I also have concerns about the paper that I feel necessary to be resolved.

Most importantly, the concept of "robustness" in GCN seems to be inconsistent throughout the paper.  Namely, the meaning of robustness in the neural network (adversarial robustness) and the SBM literature (spectral robustness) are different. This point is crucial since the paper use the spectral robustness for justification of the method, yet experiments are done on the adversarial attacks. More specifically, adversarial training methods for neural networks, e.g., adversarial attack methods [1] considered in the paper, typically make the loss function (or output of network) more persistent against the small perturbation of inputs. On the other side, the robustness for SBM models, e.g., Theorem 3 in the paper, cares more about the preservation of the original input characteristics. For illustration, an invertible neural network [2] is not necessarily robust to adversarial attacks (the first meaning of robustness) but preserves all the input characteristics (the second meaning of robustness).

I also hope the paper could have done the experiments on more datasets since there exists some evidence on the unreliability of evaluations on citation networks [3].  However, I do not think this point is critical since the paper did a great job of evaluating the robustness in various aspects and they all show consistent improvement.

Minor questions and suggestions:
- The acronyms are slightly confusing to understand at first sight, since they first appear at the equations without any information on what the letters stand for.  Something like a "variable power network (VPN)" would make the paper more pleasant to read.
- In the r-GCN framework, there might be an edge case where the powered graph is almost identical to another graph. Would there be any justification for avoiding this?
- In the r-GCN framework, the terminology distillation is slightly confusing. Was this choice of word used for making a connection to the knowledge distillation [4]? How is the knowledge distilled between graphs?

References
[1] Bojchevski and Günnemann. Adversarial attacks on node embeddings via graph poisoning. ICML 2019
[2] Jacobsen et al., i-RevNet: Deep Invertible Networks. ICLR 2018
[3] Shchur et al., Pitfalls of Graph Neural Network Evaluation, Arxiv 2018
[4] Hinton et al., Distilling the Knowledge in a Neural Network, Arxiv 2015

**Experience Assessment:**

I have read many papers in this area.

**Review Assessment: Checking Correctness Of Derivations And Theory:**

I did not assess the derivations or theory.

**Review Assessment: Checking Correctness Of Experiments:**

I assessed the sensibility of the experiments.

**Review Assessment: Thoroughness In Paper Reading:**

I read the paper at least twice and used my best judgement in assessing the paper.

---

> ### Author Response · Authors · 2019-11-12
> **Response (part 2)**
>
> Q: The acronyms are slightly confusing to understand at first sight, since they first appear at the equations without any information on what the letters stand for. Something like a "variable power network (VPN)" would make the paper more pleasant to read.
>
> A: Thank you for the suggestion! We have addressed it in the revised paper.
>
> Q: In the r-GCN framework, there might be an edge case where the powered graph is almost identical to another graph. Would there be any justification for avoiding this?
>
> A: Thank you for introducing this interesting scenario! We do not think this poses a serious problem to our framework. In fact, every powered graph is a different graph than the original graph. But since they are derived from the original graph, they inherit some important information from the original graph, both in the spatial and in the spectral domain. For example, under SBM, the leading eigenvectors of the adjacency matrix are relatively stable for powered graphs, as proven in Theorem 3. In addition, these powered graphs are also introduced in the training as a regularization term, whose influence is controlled by the regularization parameter. Therefore, even if the powered graph is almost identical to another graph,  it does not do harm to our learning performance.
>
> Q: In the r-GCN framework, the terminology distillation is slightly confusing. Was this choice of word used for making a connection to the knowledge distillation [4]? How is the knowledge distilled between graphs?
>
> A: Thank you for this comment! In the original paper [4], the knowledge is distilled between models, where knowledge is transferred from one to another. We attempted to draw an analogy to graphs, where the knowledge learned on the powered graph is transferred to the model that is applied to the original graph. We agree with the reviewer that using this term can be confusing to some readers. To improve clarity, we replaced this term with "transfer" in the revised manuscript.
>
> Thank you once again for the useful feedback! Please do not hesitate to let us know if there are further issues that we could address.

---

> ### Author Response · Authors · 2019-11-12
> **Response (part 1)**
>
> Thank you for the insightful comments and suggestions! We are pleased that you like how the paper addressed the weakness of the existing graph Laplacian operators and that we proposed a new method with theoretical justifications. We also thank you for acknowledging our thorough experimental evaluation of the proposed method. In the following, please see our detailed responses to the concerns that you have raised.
>
> Q: Most importantly, the concept of "robustness" in GCN seems to be inconsistent throughout the paper. Namely, the meaning of robustness in the neural network (adversarial robustness) and the SBM literature (spectral robustness) are different. This point is crucial since the paper use the spectral robustness for justification of the method, yet experiments are done on the adversarial attacks. More specifically, adversarial training methods for neural networks, e.g., adversarial attack methods [1] considered in the paper, typically make the loss function (or output of network) more persistent against the small perturbation of inputs. On the other side, the robustness for SBM models, e.g., Theorem 3 in the paper, cares more about the preservation of the original input characteristics. For illustration, an invertible neural network [2] is not necessarily robust to adversarial attacks (the first meaning of robustness) but preserves all the input characteristics (the second meaning of robustness).
>
> A: Thanks for raising this important issue! We apologize for not making the text as clear as we intended.  Indeed "spectral robustness" and "adversarial robustness" are two different concepts in two communities. In this work, we argue that spectral robustness is helpful to improve adversarial robustness. The mathematical explanation is that the perturbation of the leading eigenvectors of the convolutional operator is controlled by the inverse of the spectral gap. More specifically, let $\lambda_k$ and $\mathbf{\phi}_k$ be the eigenvalue and the corresponding eigenvector of the adjacency matrix $\mathbf{A}$. Then, the sensitivity of the eigenvector to the perturbation is given by:
> $$\frac{\partial \mathbf{\phi}_k}{\partial\mathbf{A}_{(ij)}}=\sum_{\ell=1,\ell\neq k}^n\frac{\phi_{\ell(i)}\phi_{k(j)}(2-\delta_{(ij)})}{\lambda_k-\lambda_\ell}\mathbf{\phi}_\ell,$$
> where $\mathbf{A}_{(ij)}$ is the $(i,j)$-th entry of $\mathbf{A}$, $\phi_{k(j)}$ is the $j$-th entry of $\mathbf{\phi}_{k}$,  $\delta_{(ij)}$ is the perturbation of the $(i,j)$-th entry of $\mathbf{A}$, and $n$ is the dimension of $\mathbf{A}$. It can be seen that the perturbation of the convolutional operator is controlled by the inverse of the spectral gap. When the spectral gaps between the leading eigenvalues (i.e., $\lambda_1$ and $\lambda_2$) and the rest of the eigenvalues are large, the perturbations of the leading eigenvectors due to the adversarial attack are also controlled. Since the eigenvalues of the graph convolutional operator decays very quickly (see Figure A.2 as an illustration), the learned representation is heavily influenced by the leading eigenvectors. Thus, by controlling the perturbations of the leading eigenvectors, one can expect to control the perturbations of the outputs.
>
> We also thank the reviewer for the interesting reference! We have discussed its relation and added it to our paper.
>
> Q: I also hope the paper could have done the experiments on more datasets since there exists some evidence on the unreliability of evaluations on citation networks [3]. However, I do not think this point is critical since the paper did a great job of evaluating the robustness in various aspects and they all show consistent improvement.
>
> A: Thank you for raising this important issue! Also, thank you for acknowledging our substantial efforts in evaluating the robustness in various aspects and demonstrating a consistent improvement of the proposed method. Indeed, By focusing on the three most common benchmark datasets, we are able to make sure that our implementation of existing methods achieves similar performance on the clean data setting. This makes it possible to reliably estimate their performance in an adversarial setting against various state-of-the-art attack strategies. We also thank the reviewer for pointing out the reference! The paper [3] points out the unreliability of evaluation on the benchmark datasets in the clean setting, but it remains to be seen whether it is also the case for the adversarial setting. By using various attack strategies, our aim is to control the variance of robustness evaluation, so to alleviate this unreliability issue as much as we can.

---

### Official Review · AnonReviewer3 · 2019-10-23
**Official Blind Review #3**

**Rating:** 3

**Review:**

In this paper, the authors study the classic GCN and proposed the new convolution operator with wider spatial scope and robust properties. The proposed models could improve the accuracy in both benign and evasion setting on synthetic, ie., SBM dataset and real world benchmark graphs. However, I have the following questions for the authors:

1. In the paper, compared to the classic GCN, the authors replace the adjacent matrix $A$ with the proposed “variable power operator”. However, the proposed “variable power operator” is very similar to “k-th order polynomials of the Laplacian”, which has been fully discussed in [1]. Could you distinguish the differences between the proposed “variable power operator” and “k-th order polynomials of the Laplacian”? And as the authors proposed to use high-order matrix  some recent models which also explores high-order matrix such as [2, 3] may also need to be selected as baseline methods.

2. As it is clearly defined in [4,5], all the five attack methods adopted in the paper are poisoning (training time) attacks methods. However, the proposed models are claimed to defense against evasion (testing) attacks. Why choose the poisoning attacks methods here? The experiment with the evasion attack method [6] is suggested to be added.

3. When the proposed models are applied to other kind of graph data, ie., social network, according to the small world theory, the "variable power adjacency matrix" would be very dense when $r>2$ with 2-layer GCN. The efficiency of the proposed might be an issue. Is it possible to add one experiment with demonstrating the running time on the real world social network?

[1] Defferrard, Michaël, Xavier Bresson, and Pierre Vandergheynst. "Convolutional neural networks on graphs with fast localized spectral filtering." Advances in neural information processing systems. 2016.
[2]Wu, Felix, et al. "Simplifying graph convolutional networks." International Conference on Machine Learning. 2019.
[3] Abu-El-Haija, Sami, et al. "Mixhop: Higher-order graph convolution architectures via sparsified neighborhood mixing."  International Conference on Machine Learning. 2019.
[4]Zügner, Daniel, and Stephan Günnemann. "Adversarial attacks on graph neural networks via meta learning." In ICLR 2019.
[5]Bojchevski, Aleksandar, and Stephan Günnemann. "Adversarial Attacks on Node Embeddings via Graph Poisoning." International Conference on Machine Learning. 2019.
[6] Dai, Hanjun, et al. "Adversarial attack on graph structured data." International Conference on Machine Learning. 2018.




**Experience Assessment:**

I have published in this field for several years.

**Review Assessment: Checking Correctness Of Derivations And Theory:**

I carefully checked the derivations and theory.

**Review Assessment: Checking Correctness Of Experiments:**

I assessed the sensibility of the experiments.

**Review Assessment: Thoroughness In Paper Reading:**

I read the paper thoroughly.

---

> ### Author Response · Authors · 2019-11-12
> **Response (part 2)**
>
> Q: As it is clearly defined in [4, 5], all the five attack methods adopted in the paper are poisoning (training time) attacks methods. However, the proposed models are claimed to defend against evasion (testing) attacks. Why choose the poisoning attacks methods here? The experiment with the evasion attack method [6] is suggested to be added.
>
> A: Thank you for this comment! The main difference between the poisoning attack and the evasion attack is whether the target model is retrained to update the parameters of the model after the attack. Since we assume the model parameters remain unchanged in our theory, we choose the evasion attack in our experimental settings. Furthermore, we aim to defend the global attack methods to be consistent with our theory, where the percentage of attacked edges indicates how the severity of the attack. All the five attack methods are very strong and state-of-the-art global attack methods that can be found in the literature and they also perform effectively under the evasion setting though they were proposed under the poisoning attack setting. Thus we choose them as attack methods for experiments. We thank the reviewer for pointing out the interesting work [6]. The method in [6] is developed for the targeted attack as opposed to the global attack. Moreover, we find that it could not finish within the given time limit even for the case where 30% nodes are selected as target nodes on smaller datasets such as Citeseer and Cora. Therefore, we only discuss its relevance in the paper.
>
> Q: When the proposed models are applied to other kind of graph data, ie., social network, according to the small world theory, the "variable power adjacency matrix" would be very dense when with 2-layer GCN. The efficiency of the proposed might be an issue. Is it possible to add one experiment with demonstrating the running time on the real world social network?
>
> A: Thank you for raising this important issue! We agree with the reviewer that the adjacent matrix could be very dense after powered several times, which is a general challenge for all high-order matrix based approaches. To alleviate this issue, we employ a simple sparsification strategy in our proposed method. Thank you for your suggestion for the additional experiment! We introduced a running time comparison experiment on the real world social network (Social circles: Facebook) [A1]. This dataset consists of 'circles' (or 'friends lists') from Facebook and becomes very dense from power 1 (4.71%) to power 2 (92.12%), thus is suitable for this scenario. The results are as follows:
> METHOD | Vanilla GCN | PowerLaplacian | IGCN(RNM) | IGCN(AR) | LNet | RGCN | SGC | MixHop | r-GCN | VPN |
> Run Time (s) | 6.02 | 6.36 | 3.33 | 7.21 | 5.56 | 18.14 | 0.231 | 11.38 | 6.73 | 7.18
> We can find that the running efficiency of our proposed method is compatible with baselines and the density doesn't affect the efficiency significantly when the dataset is not too large. Moreover, in this paper, our primary goal is to improve the robustness. We leave the solution to resolve the scalability issue as a future direction.
>
> [A1] J. McAuley and J. Leskovec. Learning to Discover Social Circles in Ego Networks. NIPS, 2012. https://snap.stanford.edu/data/ego-Facebook.html
>
> We thank the reviewer once again for the constructive feedback to improve the paper further. Please do not hesitate to let us know if there are further issues that we could address.

---

> ### Author Response · Authors · 2019-11-12
> **Response (part 1)**
>
> We thank the reviewer for raising many important points to improve the paper! We are also pleased that the reviewer acknowledges the improvement of our proposed methods for accuracy and robustness in both synthetic and real world benchmark graphs. We have added the additional baselines [2,3] in our evaluation to further substantiate our claim. We also discussed the issues and clarified the text to address the questions raised by the reviewer. Please see the detailed responses below.
>
> Q: In the paper, compared to the classic GCN, the authors replace the adjacent matrix with the proposed “variable power operator”. However, the proposed “variable power operator” is very similar to “k-th order polynomials of the Laplacian”, which has been fully discussed in [1]. Could you distinguish the differences between the proposed “variable power operator” and “k-th order polynomials of the Laplacian”? And as the authors proposed to use high-order matrix some recent models which also explores high-order matrix such as [2, 3] may also need to be selected as baseline methods.
>
> A: Thank you for this important question! There is an important difference between the proposed operator and the k-th order polynomial. If you power the graph Laplacian L to the k-th order, the resulting matrix has the same eigenvectors as of L, and only the eigenvalues are powered to the k-th order. Thus, the resulting k-th order polynomial has the same eigenvectors, so summing them up does not change the eigenvectors. However, the proposed variable power operator has radically different eigenvectors as the graph Laplacian (or it's k-th order). This is an important difference because we know (empirically) that the leading eigenvectors of a graph Laplacian are extremely sensitive to outliers under the SBM, and they often correspond to either tails or high-degree nodes (please see Fig. A.3 for an illustration). However, we have proved in Theorem 3 that the leading eigenvectors of the proposed operator can asymptotically recover the underlying community under SBM, and enjoys the "spectral gap" property. This is also a fundamental difference between our proposed method and those proposed in [2,3]. We added this discussion in Section 4.4.
>
> Thank you for the suggestion to add the comparison with baselines SGC [2] and MixHop [3]. We have also added them in our experiments. For simplicity, we provide the results on dataset Citeseer, attacked by ADW3 here and include the full results in the revision:
> METHOD | Vanilla GCN | PowerLap2 | PowerLap3 | GCN(RNM) | IGCN(AR) | LNet | RGCN | SGC | MixHop | r-GCN | VPN
> 5% | 68.8 | 69.2 | 70.0 | 63.5 | 65.1 | 60.0 | 70.3 | 70.5 | 70.4 | 71.8 | 70.6
> 10% | 68.4 | 69.3 | 69.6 | 63.1 | 63.6 | 59.6 | 70.1 | 70.1 | 69.4 | 71.2 | 70.2
> 15% | 68.9 | 69.5 | 69.8 | 62.9 | 63.2 | 59.5 | 70.3 | 70.6 | 68.7 | 71.2 | 70.6
> 20% | 68.8 | 69.4 | 69.5 | 63.5 | 63.4 | 59.1 | 69.8 | 70.3 | 67.9 | 71.1 | 70.4
> 25% | 68.8| 69.2 | 69.3| 63.6 | 63.8 | 59.3 | 69.9 | 70.2 | 67.7 | 71.2 | 70.1
> 30% | 68.78| 69.2 | 69.3| 63.6 | 63.8 | 59.3 | 69.9 | 70.2 | 67.3 | 71.2 | 70.1
> From the above, we can find that our proposed method is consistently outperforms the baselines even with [2, 3] added as baselines.

---

### Official Review · AnonReviewer1 · 2019-10-23
**Official Blind Review #1**

**Rating:** 3

**Review:**

This paper proposes a graph convolutional operator based on graph powering and applies it to GCN architecture to improve the performance and robustness. This work is mainly motivated by the paper (Graph powering and spectral robustness, Abbe et al., 2018). The authors introduce the graph powering to graph convolution neural network domain to replace the original Laplacian operator. They further propose a graph sparsification/pruning strategy on the powered adjacency matrices in order to reduce the complexity and increase the robustness against adversarial attacks. They also provide theoretical analysis to prove that the proposed powering operator and subsequent methods have some spectral properties and theoretical feasibility. However, some conclusions are limited to the ideal situations or seem subjective. Extensive experiments are conducted to show better or comparable performance in both benign and adversarial situations.

There are some concerns that need to be addressed or clarified:
A major concern is that the theoretical analyses in this paper are limited to graphs sampled from the SBM model. It is unclear how these analyses can be generalized to real graphs. Furthermore, the theorem 3 and proposition 5 are even limited to SBM model with 2 communities, which makes the analyses less convincing.

Some of the arguments in the paper might be imprecise. For example, in Section 1.1, when discuss “why not graph Laplacian?”, a small spatial scope is claimed to be problematic. Although, it is correct for the GCN (Kipf & Welling, 2017), the powered Laplacian (mentioned earlier in the same section) does have a broad spatial scope.

It would be better if the authors could provide more details about the sparsification. Specifically, how to choose the threshold (adaptively).

In the performance part of Section 4.2, the improvement of the performance by replacing Laplacian with VPN is marginal (compared with the original GCN). Furthermore, the performance of VPN is close to or sometimes worse than the baseline RGCN.

Suggestions:
In the Informative and robust low-frequency spectral signal part of Section 4.3, it would be better if the authors can clarify the experiments setting. Is it using the low-frequency part (first few eigenvectors) to recover the signal and then using the recovered signal to perform the classification task? The titles of Figure 7 and Figure 8 are a little bit confusing.


Some minor problems:
There are many typos such as: “with the presence of absence of edges”, “normalizating”, “asymptotoic”, “benigh”, “ajdacency”, “sensitve”, “adajacent”, “one of the network”, etc.

**Experience Assessment:**

I have published in this field for several years.

**Review Assessment: Checking Correctness Of Derivations And Theory:**

I did not assess the derivations or theory.

**Review Assessment: Checking Correctness Of Experiments:**

I assessed the sensibility of the experiments.

**Review Assessment: Thoroughness In Paper Reading:**

I read the paper at least twice and used my best judgement in assessing the paper.

---

> ### Author Response · Authors · 2019-11-12
> **Response (part 2)**
>
>
> Q: In the Informative and robust low-frequency spectral signal part of Section 4.3, it would be better if the authors can clarify the experiments setting. Is it using the low-frequency part (first few eigenvectors) to recover the signal and then using the recovered signal to perform the classification task? The titles of Figure 7 and Figure 8 are a little bit confusing.
>
> A: Thank you for this valuable suggestion! We have added more details about the experimental setting in Section 4.3. For the experiments in Section 4.3, we first perform eigen-decomposition of the graph convolutional operator (i.e., graph Laplacian or VPN) to obtain the Fourier modes $\Phi$. We then reconstruct the nodal features $X$ using only the k-th and k+1-th eigenvectors, i.e., $\Phi_{:,k:(k+1)}\Phi_{:,k:(k+1)}^\top X$. We then use the reconstructed features in MLP to perform the classification task in a supervised learning setting. We performed this experiments for all three datasets. We have also changed the titles of Fig. 7 and 8 to avoid the confusion.
>
> Q: There are many typos such as: “with the presence of absence of edges”, “normalizating”, “asymptotoic”, “benigh”, “ajdacency”, “sensitve”, “adajacent”, “one of the network”, etc.
>
> A: Thank you for catching these typos! We have corrected them in the revised version.
>
> We thank the reviewer once again for the very interesting and important remarks! Please do not hesitate to let us know if there are further issues that we could help clarify.

---

> ### Author Response · Authors · 2019-11-12
> **Response (part 1)**
>
> We thank the reviewer for raising the interesting and important points! We are also pleased that the reviewer acknowledges our efforts to extensively evaluate the performance of our method and demonstrate its consistent improvement in both benign and adversarial situations. Please see our detailed responses below.
>
> Q: A major concern is that the theoretical analyses in this paper are limited to graphs sampled from the SBM model. It is unclear how these analyses can be generalized to real graphs. Furthermore, the theorem 3 and proposition 5 are even limited to SBM model with 2 communities, which makes the analyses less convincing.
>
> A: Thank you for raising this important point! SBM was a classic model proposed in mathematical sociology and it has been adopted to model and analyze real-world social and biological networks. The main advantage of SBM is that it encodes the structural information that nodes belong to the same community tend to be more connected with each other. But indeed, we recognize this limitation that SBM might not be a good model for some applications. For example, models with more flexibility in the degree distributions, such as the degree-corrected SBM proposed by Karrer and Newman (2011), labeled SBM by Heimlicher (2012), or hierarchical SBM proposed by Peixoto (2017) could be more suitable. As the reviewer points out, our analysis is limited to 2 communities; extension of the theory to more than 2 communities is an interesting yet challenging open problem. However, empirically, we observe that our theory has strong implications for real-world graphs. This is evident from the frequency analysis experimental results in Fig. 7 and 8. The proposed graph convolution operator has a clear advantage in the low-frequency regime (i.e., leading eigenvectors) over the classic graph Laplacian in terms of clustering accuracy. We have added a new section 4.4 to discuss the above. Thanks again for this important remark!
>
> Q: Some of the arguments in the paper might be imprecise. For example, in Section 1.1, when discuss “why not graph Laplacian?”, a small spatial scope is claimed to be problematic. Although, it is correct for the GCN (Kipf & Welling, 2017), the powered Laplacian (mentioned earlier in the same section) does have a broad spatial scope.
>
> A: We agree with the reviewer and we apologize for not making the arguments are precise as we intended. Indeed, many recent works such as MixHop, SGC, powered Laplacian have addressed the issue of limited scope by powering the graph Laplacian directly. We have modified the text to discuss this (both in Section 1.1 and Section 4.4).
>
> Q: It would be better if the authors could provide more details about the sparsification. Specifically, how to choose the threshold (adaptively).
>
> A: Thank you for this suggestion! In our experiments, we adopt a simple sparsification strategy. For each node, we consider all its neighbors within r hops, where r is the order of the powered graph. We order these neighbors based on the Euclidean distance on the nodal features. We then set the threshold for this node such that only the first s*d nearest neighbors in the feature space are selected, where s is the sparsification rate, and d is the degree of the node in the original graph. If this node is a high-degree node (determined by how many standard deviations is its degree compared to the average), we set s to be a small number compared to the case when the node is not a high-degree node. Nevertheless, we do not observe that the robustness performance is sensitive to this parameter. In addition, one can define their own distance function that better measures proximity between nodes, such as correlation distance or cosine distance. We have added this detail in appendix section A.5(more experimental details and results).
>
> Q: In the performance part of Section 4.2, the improvement of the performance by replacing Laplacian with VPN is marginal (compared with the original GCN). Furthermore, the performance of VPN is close to or sometimes worse than the baseline RGCN.
>
> A: Thank you for this comment. Among all the evaluated methods (we also added two additional baselines, SGC and MixHop), the performance of our method is consistently ranked in the top two in the clean dataset (please see Table I for the updates). In the adversarial setting, our extensive experiments indicate that our method is more robust than the baselines for the majority of the time. Therefore, the key strength of the proposed method is that it can *simultaneously* improve accuracies in both the clean and the adversarial settings, and it is this combination that makes our method unique.

---

### Author Response · Authors · 2019-11-13
**Revision uploaded**

Dear Reviewers,

Thank you for your effort in reviewing the manuscript! Your comments and remarks have been very helpful to improve the quality of the paper.

We have made three major additions to the manuscript:

1) We performed additional experiments with baselines SGC and MixHop in both the clean data setting and the adversarial setting against five strong attack strategies.
2) We added a section (Section 4.4) to discuss the important issues regarding our methods, including (i) the difference between our method and those in the literature that are based on directly powered graph Laplacian or k-th order polynomials; (ii) the relation between adversarial robustness and spectral robustness; and (iii) the limitation of the present analysis to stochastic block model.
3) We also added an experiment to evaluate the runtime of the proposed method on a social network dataset (Social circles: Facebook).

Please find the detailed responses below. We thank you again for your constructive feedback. Please do not hesitate to let us know if there are further issues that we could help clarify.

---

### Decision · Program_Chairs · 2019-12-19

**Decision:**

Reject

**Comment:**

The paper identifies the limitation of graph neural networks and proposed new variants of graph neural works. However, the reviewers feel that the theory of the paper have some problems:
1. A major concern is that the theoretical analyses in this paper are limited to graphs sampled from the SBM model. It is unclear how these analyses can be generalized to real graphs.
2. The robustness definition is inconsistent.
Furthermore, more extensive experiments on more datasets will also be helpful.